# WeatherSyn: An Instruction Tuning MLLM For Weather Forecasting Report Generation

**Zinan Zheng** [1]  **Yang Liu** [* 2]  **Nuo Chen** [* 3]  **Juepeng Zheng** [4 5]  **Hong Cheng** [2]  **Jia Li** [1]

## Abstract

Accurate weather forecast reporting enables individuals and communities to better plan daily activities and agricultural operations. However, the current reporting process primarily relies on manual analysis of multi-source data, which leads to information overload and reduced efficiency. With the development of multimodal large language models (MLLMs), leveraging data-driven models to analyze and generate reports in the weather forecasting domain remains largely underexplored. In this work, we propose the Weather Forecasting Report (WFR) task and construct the first instruction-tuning dataset for this task, named **WSInstruct**, which covers 31 cities in America and 8 weather aspects. Based on this corpus, we develop the first model, **WeatherSyn**, specialized in generating weather forecast reports. Evaluation across multiple metrics on our dataset shows that **WeatherSyn** consistently outperforms leading closed-source MLLMs, particularly on structurally complex weather aspects. We further analyze its performance across diverse geographic regions and weather aspects. **WeatherSyn** demonstrates strong transferability across different regions, highlighting its zero-shot generalization capability. **WeatherSyn** offers valuable insight for developing MLLMs specialized in weather report generation. Codes are available at https://github.com/compasszzn/WeatherSyn.

## 1. Introduction

The development of artificial intelligence has driven progress across numerous scientific domains, including crystallography (Cao et al., 2025a;b), physics (Liu et al., 2024; Zheng et al., 2024; Zhao et al., 2026), and meteorology (Ukhurebor et al., 2022; Zheng et al., 2025). Weather forecasting provides critical support for social resilience by providing accurate and timely predictions of atmospheric conditions. Precise forecasts of variables such as temperature, humidity, precipitation, and cloud cover facilitate effective planning in daily operations, agriculture (Liu et al., 2025a;b; Li et al., 2025), and transportation. The traditional weather forecasting report workflow is illustrated in Figure 1. Weather experts rely on physics-based Numerical Weather Prediction (NWP) models, which solve discretized thermodynamic and fluid dynamical equations (Nathaniel et al., 2024), to generate forecast fields from initial conditions. Meteorologists then analyze the initial conditions and forecast states, integrating observational data through collaborative discussions, to produce the final forecast reports. However, this process faces significant challenges: information overload from hundreds of variables across diverse data sources limits efficiency. Additionally, the incorporation of subjective judgments will lead to inconsistencies across reports. Recent advances in multimodal large language models (MLLMs) offer a promising avenue to address these issues. Their ability to interpret and integrate multi-variable imagery data can assist forecasters by reducing workload and minimizing subjective biases, thereby enhancing the consistency and comprehensiveness of weather reporting.

MLLMs have recently been applied to weather captioning tasks. WeatherQA (Ma et al., 2024) introduces the first multimodal dataset for extreme event weather report generation. OmniEarth-Bench (Wang et al., 2025) includes captioning and open-ended analysis tasks for atmospheric conditions, while Omni-Weather (Zhou et al., 2025) focuses on radar-based precipitation nowcasting report generation. Despite these efforts, existing work remains far from addressing the broader problem of general weather forecast report generation. This gap stems primarily from two challenges: the scarcity of publicly available datasets with paired visual data and textual weather reports, and the limited ability of current MLLMs to perform domain-specific reasoning over complex, multi-variable meteorological inputs.

To address this gap, we introduce Weather Forecasting Report (WFR) as a new task within the field of weather

[1]Hong Kong University of Science and Technology (Guangzhou) [2]The Chinese University of Hong Kong [3]HY Foundation Model Team, Tencent [4]Sun Yat-Sen University [5]National Supercomputing Center in Shenzhen. Correspondence to: Yang Liu <yliuweather@gmail.com>, Nuo Chen <chennuo26@gmail.com>.

*Proceedings of the 43rd International Conference on Machine Learning*, Seoul, South Korea. PMLR 306, 2026. Copyright 2026 by the author(s).

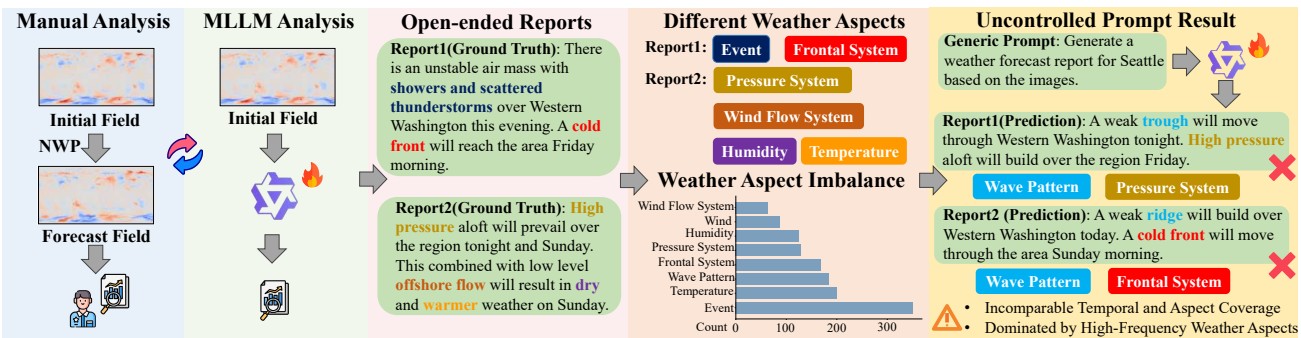

*Figure 1.* The pipeline of the weather forecast reporting process and the challenges in open-ended weather forecast report generation

forecasting. WFR is designed to directly generate human-readable weather reports from the initial atmospheric conditions at a given time $t$ and position $p$. By eliminating dependence on intermediate numerical forecasts, this approach streamlines the forecasting process and facilitates more accessible, timely information to the public.

In this paper, we construct the first Weather Forecast Report (WFR) instruction-tuning dataset, named **WSInstruct**. The dataset pairs city-level variable heat maps converted from the Earth Reanalysis 5 (ERA5) dataset (Hersbach et al., 2020) as visual inputs with expert-written weather forecast reports as textual ground truth. **WSInstruct** covers the report of 31 cities in the United States and spans 8 different weather aspects. As illustrated in Figure 1, due to the open-ended nature of real-world weather reports, different reports often emphasize different weather aspects. Training models without explicit aspect constraints leads to uncontrolled and incomparable generations across both temporal scope and aspect coverage. Moreover, the generated reports tend to become homogeneous, with content dominated by high-frequency weather aspects observed in the training data. To address these challenges, we introduce aspect-controlled prompting in **WSInstruct** to enable fine-grained and controllable weather report generation.

Based on this corpus, we propose the first open-source MLLM specialized in weather report generation. Our approach proceeds in three stages. First, we perform supervised fine-tuning (SFT) of the open-source Qwen3-VL-8B model using **WSInstruct**, yielding the **WeatherSyn**. Second, to improve the lexical diversity of the generated reports, we apply rejection sampling on the SFT model to produce outputs that are lexically diverse while remaining factually correct, which are incorporated into an augmented dataset. We then fine-tune Qwen3-VL-8B on this augmented corpus, which we term Rejection Sampling Fine-Tuning (RFT). Finally, we apply Direct Preference Optimization (DPO) alignment to refine the model's style and factuality, ensuring that generated forecasts are faithful to expert descriptions. Through extensive experiments on the **WSInstruct**, we find that

- **WeatherSyn** significantly outperforms leading closed-source and specialized weather MLLMs across multiple evaluation metrics, including Claude-3.7-Sonnet, GPT-5-Nano, and WeatherQA, producing weather reports that are both meteorologically accurate and human-readable.

- Generalization experiments demonstrate that, when trained on data from several cities, our model generates reports for unseen cities in a zero-shot setting that still surpass the few-shot performance of close-source models, highlighting its strong transferability.

- We further conduct a comprehensive analysis of the effects of training data scale, the results of different area, weather aspect and forecast time step. The results provide practical insights into model optimization and performance improvement.

## 2. Problem Definition

Consider a weather forecast report dataset $\mathcal{D} = \{\mathbf{Q}_t^p, \mathcal{I}_t^p, \mathbf{R}_{t,t+d}^p\}$, where $\mathbf{Q}_t^p$ denotes the query for specific time $t$ and position $p$, with $p$ simplified to denote a city in our setting. $\mathcal{I}_t^p = \{\mathbf{I}_t^{p1}, \mathbf{I}_t^{p2}, ..., \mathbf{I}_t^{pi}\}_{i=1}^{N_v}$ represents a set of $N_v$ variable-specific weather heatmaps (e.g., temperature, precipitation, wind), which reflect meteorological conditions at time $t$ and city $p$. $\mathbf{R}_{t,t+d}^p$ represents the ground-truth forecast report issued at time $t$ for city $p$, which describes the predicted weather conditions over the future period from day $t$ to day $t + d$. The task for weather forecast report generation is to train a model that generates the forecast report $\hat{\mathbf{R}}_{t,t+d}^p$ conditioned on both the image set $\mathcal{I}_t^p$ and the instruction $\mathbf{Q}_t^p$.

## 3. Methodology

We aim to develop a general weather report generation model capable of producing textual forecast synopses comparable to those written by experts. An overview of our

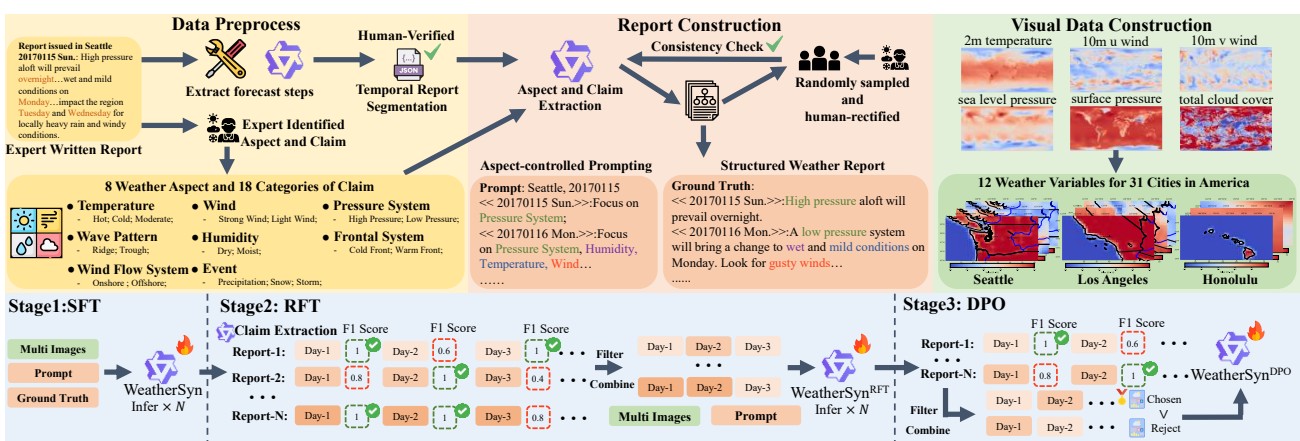

*Figure 2.* Construction of the **WSInstruct** weather forecast report dataset and the three-stage training strategy

*Table 1.* Statistics of our corpus, including total image sets ($\mathcal{I}$) and report $\mathbf{R}$.

| | Data | Year | Sum. |
|---|---|---|---|
| **WSInstruct** | Total $\mathcal{I} - \mathbf{R}$ | 2017-2021 | 6344 |
| **WSInstruct-RFT** | Total $\mathcal{I}$
Total $\mathbf{R}$ | 2017-2020 | 20412 |
| **WSInstruct-DPO** | Total $\mathcal{I} - (\mathcal{Y}_w, \mathcal{Y}_l)$ | 2021 | 1241 |
| **Test Set** | Total $\mathcal{I} - \mathbf{R}$ | 2022 | 1292 |

framework is illustrated in Figure 2. We adopt a three-stage training strategy: (1) Supervised Fine-Tuning (SFT). In the first stage, we construct the **WSInstruct** and fine-tune the base model Qwen3-VL-8B with it, yielding **WeatherSyn** . We denote the dataset and the resulting model as $\mathcal{D}$ and $\mathcal{M}$, respectively. This stage enables the model to learn to interpret visual inputs and reason about future weather conditions. (2) Rejection Sampling Fine-Tuning (RFT). RFT has been widely used to augment data (Chen et al., 2024a;b). In the second stage, we enhance the original dataset via rejection sampling to construct an augmented version, denoted as $\mathcal{D}_{RFT}$, which encourages better alignment between visual inputs and lexically diverse yet factually accurate descriptions. This augmented dataset is then used to fine-tune the base Qwen-3-VL-8B model, obtaining **WeatherSyn-RFT**, denoted as $\mathcal{M}_{RFT}$. and (3) Direct Preference Optimization (DPO). In the final stage, we construct a preference dataset **WSInstruct-DPO**, denoted as $\mathcal{D}_{DPO}$, and apply Direct Preference Optimization (DPO) to further refine the model $\mathcal{M}_{RFT}$. This process yields **WeatherSyn-DPO**, denoted as $\mathcal{M}_{DPO}$, which generates higher-quality forecasts that align more closely with human preferences and expert judgment. The statistics of the data used at each stage are summarized in Table 1, with further details provided in Appendix A.2. In the following sections, we provide detailed descriptions of the training strategies and dataset construction for each stage.

### 3.1. Training Strategy

**Stage 1 & Stage 2** We adopt a mixed-task instruction tuning strategy, in which the model is trained simultaneously on data from 31 different cities. This approach aims to equip the model with the ability to generate diverse forecast reports tailored to various regional contexts, which is defined as follows:

$$\mathcal{L}_{\text{VLM}} = -\mathbb{E}_{(\mathbf{Q}_t^p, \mathcal{I}_t^p, \{\mathbf{R}_{t,t+d}^{p,i}\}_{i=1}^N) \sim \mathcal{D}} \left[ \sum_{i=1}^N \log p_\theta(\mathbf{R}_{t,t+d}^{p,i} \mid \mathcal{I}_t^p, \mathbf{Q}_t^p) \right]. \quad (1)$$

where $p$ denotes the cities, $t$ denotes the time, $\theta$ are the parameters of the model and $N$ denotes the number of reports for each sample. In stage 1, $N$ is fixed to 1, whereas in stage 2, the value of $N$ depends on the candidate votes for the data corresponding to city $p$ and time $t$, as described in Section 3.2.3. The optimization process trains the model to align visual meteorological inputs with corresponding textual forecasts, guiding it to effectively follow the given instructions, yielding the model $\mathcal{M}$ and $\mathcal{M}_{RFT}$, respectively.

**Stage 3** In stage 3, we employ DPO to encourage the model to learn from positive examples with the highest F1 scores while penalizing negative examples with the lowest F1 scores, thereby guiding the model to prefer generation strategies that consistently lead to more accurate and reliable predictions. We collect 1241 preference samples named $\mathcal{D}_{DPO}$ for our DPO training, and the training objective for DPO is defined as follows:

$$\mathcal{L}_{\text{DPO}}(\pi_\theta, \pi_{ref}) = -\mathbb{E}_{(x, \mathcal{Y}_w, \mathcal{Y}_l) \sim \mathcal{D}_{DPO}} \left[ \log \phi\left(\beta \log \frac{\pi_\theta(\mathcal{Y}_w|x)}{\pi_{\text{ref}}(\mathcal{Y}_w|x)} - \beta \log \frac{\pi_\theta(\mathcal{Y}_l|x)}{\pi_{\text{ref}}(\mathcal{Y}_l|x)}\right) \right]. \quad (2)$$

where $\pi_{\text{ref}}$ refers to the reference model $\mathcal{M}_{RFT}$, and the fully trained model $\pi_\theta$ is denoted as $\mathcal{M}_{DPO}$.

## 3.2. Dataset Collection

To achieve this goal, the main challenge lies in the lack of data. To address this challenge, we first curate **WSInstruct**, a multimodal weather report instruction-tuning dataset. It consists of (1) heatmap visualizations of meteorological variables around each city, and (2) corresponding expert-written weather reports forecasting the next four days. This section details the dataset construction, covering report collection and visual input generation.

### 3.2.1. TEXTUAL DATA

We collect weather reports from 31 cities, with detailed statistics in Appendix A. To ensure diversity, we keep at most one report per city per day and filter out temporally adjacent ones.

The collected reports are open-ended, varying in forecast horizon (single or multiple days) and emphasized meteorological aspects (e.g., temperature vs. wind). To address this heterogeneity, we convert each textual report into a structured representation and employ aspect-controlled prompting to steer generation, as described next.

- **Temporal Report Segmentation** To transform unstructured forecast reports into temporally structured samples, we segment each report into four consecutive daily forecasts (see first part of Figure 2). First, we identify relative day expressions (e.g., "Today", weekdays) using regular expressions, then convert them to absolute calendar dates with `dateparser`, using the report timestamp as reference. This anchors all forecast mentions to explicit dates, yielding a consistent timeline. We retain only reports that cover at least three of the first four consecutive days, balancing multi-day coverage with data availability. Segmentation into daily forecasts is performed by Qwen-2.5-72B, guided by the extracted timeline. To ensure quality, we apply a two-step verification: (1) a `regular expression` check to verify that the extracted content originates from the original report and that the corresponding dates match those obtained via `dateparser`; (2) mismatched samples are manually corrected by annotators. Prompts and an example are provided in the Appendix F.1.

- **Weather Claim Extraction and Consistency Check** The high variability in weather report content makes it challenging to evaluate model outputs against references. To enable fine-grained evaluation. Experts define eight core weather aspects from the metadata: *temperature*, *wind*, *humidity*, *frontal systems*, *pressure systems*, *wave patterns*, *wind flow systems*, and *events*. For each aspect, experts define fine-grained claim categories (Figure 2). Each claim category is accompanied by a set of reference keywords and example phrases, which together form a standardized annotation protocol detailed in Appendix Table 13.

Using this annotation protocol, we instruct Qwen-2.5-72B to simultaneously extract aspect and claim categories from the reports. To address potential mismatches between the extracted aspects and claims, we map each claim back to its corresponding aspect according to the protocol. If the mapped aspect conflicts with the initially extracted aspect, the extraction step is repeated. This process aims to improve model extraction consistency.

To validate extraction quality, domain-trained annotators manually reviewed and corrected the model's outputs for 1,292 reports from the test set. Using these human-corrected annotations as ground truth, the model achieves an extraction F1 score of 94%, confirming its reliability for this few-shot classification task. The evaluation details are in Appendix C.1; full prompts and an example are provided in Appendix F.2.

- **Aspect-controlled Prompting and Report Construction** To align model output with ground-truth aspects for evaluation, we employ aspect-controlled prompting. This method explicitly specifies the target discussion aspects for each forecast day within the prompt. Specifically, we append an aspect constraint after each date marker (e.g., `«date, weekday» Report:\n## Focus on: Temperature, Humidity`). The generated reports follow a consistent structured format `«date, weekday» Report:\n{Report}`, allowing reliable extraction of day-specific content via regular expressions.

During inference, aspect constraints are provided to both our method and the baselines. These constraints specify only the discussion aspect (e.g., temperature, wind) rather than any target values, acting as a conditioning signal. While models are capable of generating reports that cover all aspects, the explicit aspect list aligns the output with the ground-truth discussion structure, enabling consistent aspect-level comparison. This setting standardizes the discussion scope across forecast days, prevents discrepancies from missing or extraneous aspects, and provides a controllable mechanism for focusing reports on desired topics.

### 3.2.2. VISUAL DATA

To construct the visual input, we select 12 single-level variables from the ERA5 reanalysis dataset to generate regional variable heatmaps (see Appendix Table 9 for details). The dataset provides hourly data at a $0.25°$ spatial resolution. Instead of using global fields directly, we extract data for the geographic region surrounding each target city. This localized extraction ensures the visual inputs reflect the relevant meteorological context, improving report relevance. We use data from 2017–2021 for foundation training and reserve 2022 for testing.

### 3.2.3. DATA AUGMENTATION

To mitigate the scarcity of lexically diverse yet factually consistent descriptions in the textual data, we employ a step-level rejection sampling strategy to enhance the lexical variety of generated reports. This encourages the model to learn underlying weather phenomena rather than surface-level phrasing. Specifically, we sample 40 reports for each instance in a 2017–2020 subset of $\mathcal{D}$ using the Stage-1 $\mathcal{M}$ with a temperature of 0.9 (the 2021 subset is reserved for subsequent DPO training). For each generated report, we use Qwen-2.5-72B to extract claim categories per forecast day and compute the F1 score following Section 3.2.1. Since full reports rarely achieve an F1 score of 1, we retain daily sub-reports with a step-level F1 score of 1.

**Diverse Report Selection** From the filtered step-level reports, we sample a subset that ensures factual correctness while maximizing lexical diversity. We assume that reference reports exhibiting the greatest distance from the ground truth report can be considered as diverse candidates. Based on this assumption, we adopt four distance-based strategies to measure diversity: (1) edit distance, (2) TF-IDF similarity(Bafna et al., 2016), (3) Jaccard similarity, and (4) cosine similarity computed using Sentence-BERT(Reimers & Gurevych, 2019) embeddings. We then select the most diverse sub-reports according to these metrics and randomly assemble them into candidate full reports for retraining.

We augment the $\mathcal{D}$ (2017–2020 subset) with these candidate full reports to form the final $\mathcal{D}_{RFT}$. This dataset ensures high factual correctness while introducing substantial lexical variation for each visual input (e.g., "temperatures rebound" vs. "increasing temperatures"). Such phrasing diversity, which captures the same underlying meteorological phenomena in different expressions, is crucial for strengthening the image-text alignment capability of MLLMs.

### 3.2.4. PREFERENCE DATA SELECTION

During training stage 3, we employ Direct Preference Optimization (DPO) to encourage the model to generate correct forecasts. DPO is applied to the RFT-trained model $\mathcal{M}_{RFT}$, further aligning it with human-preferred outputs using a training corpus drawn from a similar domain as the SFT data. Specifically, DPO leverages input pairs labeled as $(\mathcal{Y}_w, \mathcal{Y}_l)$, where $\mathcal{Y}_w$ and $\mathcal{Y}_l$ represent the preferred and less preferred reports, respectively.

We construct these pairs from model-generated candidates. For each input in the 2021 subset of $\mathcal{D}$, we sample 40 reports from $\mathcal{M}_{RFT}$ and compute their step-level F1 scores following Section 3.2.3. The preferred report $\mathcal{Y}_w$ is assembled from daily sub-reports that achieve a step-level F1 of 1, while the less preferred report $\mathcal{Y}_l$ is formed from sub-reports with the lowest step-level scores.

## 4. Experiments

**Baselines** We evaluate several state-of-the-art closed-source MLLMs, including **GPT-4.1-Mini**, **GPT-5-Nano**,**GPT-5.2-Thinking**, **Gemini-2.5-Flash**, **Gemini-3 Pro Preview**, and **Claude-3.7-Sonnet (20250219)**. Due to the Gemini-2.5-Flash API being limited to processing a maximum of 16 images, we restrict it to the zero-shot setting. For other models, we adopt the few-shot prompt following in previous work (Ma et al., 2024). Each prompt includes the meteorological variable heatmaps, explanations of each parameter, the data timestamp, step-by-step instructions and weather aspect discussion point to guide the analysis before report generation. To enhance the relevance of the example, we retrieve few-shot samples from the same calendar month in previous years (2017–2021) for each test instance. Full prompting and implementation details are provided in Appendices F.4.

We also include comparisons with specialized meteorological models: **WeatherQA** (Ma et al., 2024) and **OmniEarth** (Wang et al., 2025). To ensure a fair comparison, we also fine-tune Qwen3-VL-8B on their publicly released data. Because these datasets are mainly classification tasks, the resulting models lack instruction-following capability for report generation. To mitigate this, we conduct an additional fine-tuning stage on 100 samples from our training set. **Implementation details** are provided in Appendices D.

**Evaluation Metric** To comprehensively evaluate the correctness of generated reports, we employ five different kinds of evaluation methods: **reference-based**, **automatic claim-based**, **human-refined claim-based**, **LLM-based**, **expert-based**.

- **Reference Evaluation** We evaluate generated reports at each forecast time step using BLEU-1, ROUGE-L and METEOR, which measure lexical relevance and structural similarity against reference texts. These metrics quantify lexical overlap and provide an overall measure of textual similarity.

- **Automatic Claim Evaluation** Claims and their corresponding aspects are automatically extracted from each report using Qwen-2.5-72B, guided by our standardized annotation protocol. To verify the reliability of this extraction, we perform a consistency check against human annotations (see Section 3.2.1).

  Given the imbalanced distribution of claim categories (e.g., within the Frontal System aspect, cold fronts account for 81% of instances), standard accuracy can be misleading. To address this, we report weighted precision and weighted F1 scores, which compute metrics per class and aggregate according to class frequency. Detailed formulations are provided in Appendix C.2.

*Table 2.* Comprehensive evaluation results of all methods on the **WSInstruct** test set across five evaluation methods.

| Algorithms | Reference Evaluation | | | Automatic Claim Evaluation | | Human-refined Claim Evaluation | | LLM Evaluation | | Expert Evaluation | |
|---|---|---|---|---|---|---|---|---|---|---|---|
| | BLEU-1 | ROUGE-L | METEOR | Precision | F1 Score | Precision | F1 Score | Fact.Cons. | Summ.Qual. | Fact.Cons. | Summ.Qual. |
| *Closed-source MLLMs* | | | | | | | | | | | |
| GPT-4.1-Mini (2-shot) | 0.20 | 0.15 | 0.13 | 0.49 | 0.49 | 0.49 | 0.48 | 0.01 | 0.01 | 0.01 | 0.01 |
| GPT-5-Nano (2-shot) | 0.16 | 0.13 | 0.10 | 0.50 | 0.47 | 0.50 | 0.48 | 0.03 | 0.03 | 0.02 | 0.02 |
| GPT-5.2 Thinking (2-shot) | 0.12 | 0.12 | 0.11 | 0.51 | 0.49 | 0.52 | 0.49 | 0.06 | 0.07 | 0.07 | 0.05 |
| Gemini-2.5-Flash (zero-shot) | 0.07 | 0.11 | 0.11 | 0.51 | 0.44 | 0.51 | 0.44 | 0.02 | 0.01 | 0.01 | 0.01 |
| Gemini-3 Pro Preview (2-shot) | 0.37 | 0.28 | 0.23 | 0.63 | 0.60 | 0.63 | 0.60 | 0.24 | 0.27 | 0.29 | 0.31 |
| Claude-3.7-Sonnet (2-shot) | 0.09 | 0.10 | 0.17 | 0.53 | 0.51 | 0.52 | 0.50 | 0.02 | 0.01 | 0.01 | 0.01 |
| *Open-source MLLMs* (with Qwen3-VL-8B) | | | | | | | | | | | |
| WeatherQA | 0.19 | 0.15 | 0.14 | 0.50 | 0.36 | 0.51 | 0.36 | 0.01 | 0.01 | 0.01 | 0.01 |
| OmniEarth | 0.17 | 0.17 | 0.14 | 0.51 | 0.40 | 0.51 | 0.40 | 0.01 | 0.01 | 0.01 | 0.01 |
| **WeatherSyn** | 0.43 | 0.31 | 0.25 | 0.60 | 0.55 | 0.60 | 0.54 | 0.11 | 0.11 | 0.11 | 0.12 |
| **WeatherSyn-RFT** | 0.43 | 0.31 | 0.25 | 0.62 | 0.59 | 0.61 | 0.59 | 0.16 | 0.15 | 0.17 | 0.15 |
| **WeatherSyn-DPO** | **0.44** | 0.32 | 0.25 | 0.62 | 0.59 | 0.63 | 0.59 | 0.33 | 0.32 | 0.29 | 0.30 |

*Table 3.* Weighted F1 scores results of all methods on the **WSInstruct** test set across 8 different weather aspect

| Algorithms | Temperature | Wind | Humidity | Frontal System | Pressure System | Wave Pattern | Wind Flow System | Event | **Average** |
|---|---|---|---|---|---|---|---|---|---|
| *Closed-source MLLMs* | | | | *Weighted F1 Score* | | | | | |
| GPT-4.1-Mini (2-shot) | 0.36 | 0.53 | 0.41 | 0.25 | 0.54 | 0.41 | 0.52 | 0.29 | 0.49 |
| GPT-5-Nano (2-shot) | 0.38 | 0.54 | 0.50 | 0.40 | 0.58 | 0.45 | 0.51 | 0.47 | 0.47 |
| GPT-5.2 Thinking (2-shot) | 0.41 | 0.54 | 0.51 | 0.32 | 0.55 | 0.50 | 0.59 | 0.57 | 0.49 |
| Gemini-2.5-Flash (zero-shot) | 0.34 | 0.54 | 0.50 | 0.26 | 0.52 | 0.48 | 0.57 | 0.37 | 0.44 |
| Gemini-3 Pro Preview (2-shot) | 0.50 | 0.63 | 0.60 | 0.48 | 0.68 | 0.60 | 0.64 | 0.69 | 0.60 |
| Claude-3.7-Sonnet (2-shot) | 0.41 | 0.56 | 0.55 | 0.26 | 0.58 | 0.53 | 0.67 | 0.59 | 0.51 |
| *Open-source MLLMs*(with Qwen3-VL-8B) | | | | | | | | | |
| WeatherQA | 0.30 | 0.54 | 0.32 | 0.22 | 0.50 | 0.30 | 0.36 | 0.36 | 0.36 |
| OmniEarth | 0.32 | 0.49 | 0.34 | 0.28 | 0.53 | 0.36 | 0.36 | 0.53 | 0.40 |
| **WeatherSyn** | 0.42 | 0.61 | 0.51 | 0.42 | 0.70 | 0.56 | 0.59 | 0.64 | 0.55 |
| **WeatherSyn-RFT** | 0.43 | 0.62 | 0.63 | 0.43 | 0.70 | 0.64 | 0.72 | 0.66 | 0.59 |
| **WeatherSyn-DPO** | 0.42 | 0.64 | 0.60 | 0.36 | 0.72 | 0.65 | 0.70 | 0.67 | 0.59 |

- **Human-refined Claim Evaluation:** We also utilize Llama-3.3-70B to extract claims and compare its outputs with those from Qwen-2.5-72B. For generated reports with conflicting claims, human annotators manually review and correct the extractions. The corrected results are then evaluated using weighted precision and weighted F1 Score, following the same protocol as the automatic claim-based evaluation.

- **LLM Evaluation:** We randomly select 144 reports from the test set and evaluate them on two criteria: factual consistency (Fact.Cons.) and summarization quality (Summ.Qual.). To improve robustness, we employ four leading closed-source LLMs: GPT-5 (2025-08-07), Gemini-2.5-Flash, Claude-3.7-Sonnet (20250219), and DeepSeek-Chat as automatic judges. Following previous work (Chen et al., 2023b), instead of asking LLMs to assign absolute scores, which can suffer from calibration inconsistency, we adopt a ranking-based protocol: each judge ranks the generated reports according to each criterion. This pairwise approach yields more stable and discriminative comparisons. We report the percentage of reports ranked first (Top-1 rate) aggregated across all LLM judges. Prompts are provided in Appendix F.3.

- **Expert Evaluation:** To address potential shortcomings in LLM evaluation, domain experts review and correct the LLM-generated rankings. During annotation, the three experts are permitted to consult the original reference

texts. If at least one expert adjusts its Top-1 ranking status, the modified result is adopted. In cases of disagreement among multiple experts, the final Top-1 status is decided by majority vote.

### 4.1. Overall Performance

The overall performance is summarized in Table 2. We observe that **WeatherSyn** consistently outperforms all baseline methods across all evaluation metrics. RFT yields noticeable improvements for **WeatherSyn** while DPO further enhances the performance. Among the baselines, open-source baselines achieve higher scores on reference-based metrics. For example, WeatherQA and Claude-3.7-Sonnet reach BLEU-1/ROUGE-L scores of 0.19/0.15 and 0.09/0.10, respectively. In contrast, under the other four evaluations, closed-source baselines show better performance. For instance, WeatherQA achieves a weighted F1 score of 0.36 and a Fact.Cons. score of 0.01, whereas GPT-5-Nano attains corresponding scores of 0.48 and 0.06. This divergence indicates that surface similarity reference-based metrics are inadequate for assessing open-ended weather report generation. The results indicate that while current specialized meteorological models still struggle to generate coherent and factually consistent weather forecast reports.

Finally, the discrepancy between the automatic claim-based and human-refined claim-based evaluations remains within

1%, validating the reliability of the automatic claim-based method for claim extraction and classification. Considering reliability, ease of implementation, and support for fine-grained analysis of the output results, we adopt the automatic claim-based F1 score as the primary evaluation metric in subsequent experiments.

### 4.2. Weather Aspect Analysis

The test set results across eight weather aspects are summarized in Table 3. We observe that our model consistently surpasses all leading closed-source MLLMs in every aspect. Notably, improvements are most pronounced for structurally complex aspects (e.g., *Pressure System* and *Wave Pattern*), where gains exceed 5%. These aspects demand an accurate interpretation of meteorological patterns beyond lexical matching, indicating that our model captures domain-specific semantics more effectively. For simpler aspects such as *Temperature* and *Wind*, improvements are more moderate.

We observe noticeable and consistent performance improvements across most aspects when applying RFT. In particular, RFT substantially enhances the F1-Score on *Humidity*, *Wave Pattern*, and *Wind Flow System*, demonstrating the effectiveness of data augmentation. The benefits of DPO are limited and aspect-dependent, yielding improvements for some aspects while leading to performance degradation for others.

### 4.3. Ablation Study of Aspect Control

To investigate the effectiveness of aspect control, we conduct an ablation study where aspect control is applied at different stages. We report the hit rate in Table 4, defined as the proportion of generated aspects that match those in the ground-truth reports. The count of each aspect in the training set is shown in Table 7.

Without explicit guidance, the model exhibits a pronounced frequency bias. When trained and tested without aspect control, it achieves higher hit rates on aspects that appear more frequently in the training data (e.g., frontal system) while overlooking less frequent but meteorologically important ones (e.g., wave patterns). A similar bias is observed when aspect control is applied only during training, where the model attains a higher hit rate on the most frequent aspect (event) in the training data.

### 4.4. Forecasting Temporal Performance Analysis

Figure 3 shows the weighted F1 scores across forecast horizons, where indices 1–4 correspond to Day 1 to Day 4 predictions. All models exhibit a consistent performance decline as the forecast horizon extends. For example, in the *temperature* aspect, Gemini-2.5-Flash drops from 0.39

to 0.35 and GPT-4.1-Mini from 0.37 to 0.34. This trend reflects the growing uncertainty inherent in longer-range weather forecasting, which propagates to the generation stage and lowers output quality.

Across all three aspects, our model maintains the highest performance over time. Compared to the SFT-only variant, RFT substantially mitigates temporal degradation, especially for the *Frontal System* and *Event* aspects where baseline models show steeper declines. This indicates that RFT enhances the model's robustness to long-range temporal uncertainty.

### 4.5. Data Volume Analysis

We examine the impact of data volume and diversity on model performance by varying the maximum number of reports retained per case during RFT. As shown in Figure 4, performance consistently improves as the training corpus expands. Specifically, when the number of reports per question increases from 1 to 4, the *Wave Pattern* and *Wind Flow System* aspects achieve gains of 0.15 and 0.17, respectively. These results indicate that greater lexical diversity in the training data enhances the model's ability to align meteorological visual inputs with their textual descriptions.

### 4.6. Generalization Analysis

To assess the generalization ability of the RFT-tuned model, we partition U.S. cities into four geographic regions: Northwest, Southwest, Northeast, and Southeast. Detailed city to region mappings are provided in Appendix A.5. We then design three cross-region training–testing splits to evaluate generalization under both geographically proximate and distant conditions. The resulting F1 scores are summarized in Figure 5.

**Geographically Proximate Generalization** We train a model using data from a random half of the cities in each of the four regions and evaluate it on the remaining cities within the same region. In this setting, training and test cities are geographically proximate and share similar regional climate characteristics. As shown in Figure 5(A), under a zero-shot evaluation, our model consistently outperforms the two-shot results of both Claude-3-7-Sonnet and GPT-5-Nano.

**Geographically Distant Generalization** To assess model robustness under more challenging out-of-domain conditions, we design two geographically distant generalization experiments: (1) training on northern U.S. cities and testing on southern ones, and (2) training on eastern U.S. cities and testing on western ones. These splits involve substantial differences in climate patterns and weather dynamics, posing a harder challenge than geographically proximate

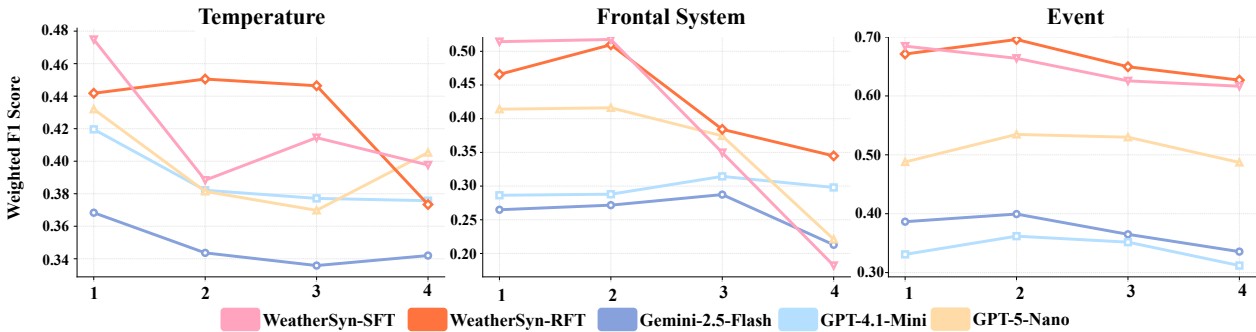

*Figure 3.* Weighted F1 scores of generated weather reports across different forecast days (Day 1–Day 4) for three weather aspects.

*Table 4.* Ablation study for aspect control in training and testing stage. We report hit rate across 8 aspects, defined as the proportion of generated aspects that align with those present in the ground-truth reports

| Train | Test | Temperature | Wind | Humidity | Frontal System | Pressure System | Wave Pattern | Wind Flow System | Event | **Average** |
|---|---|---|---|---|---|---|---|---|---|---|
| ✗ | ✗ | 0.02 | 0.01 | 0.02 | 0.43 | 0.46 | 0.07 | 0 | 0.30 | 0.16 |
| ✓ | ✗ | 0.01 | 0.04 | 0.01 | 0.01 | 0.01 | 0 | 0 | 0.88 | 0.12 |
| ✓ | ✓ | 0.92 | 0.97 | 0.92 | 0.99 | 0.98 | 0.89 | 0.90 | 0.98 | 0.94 |

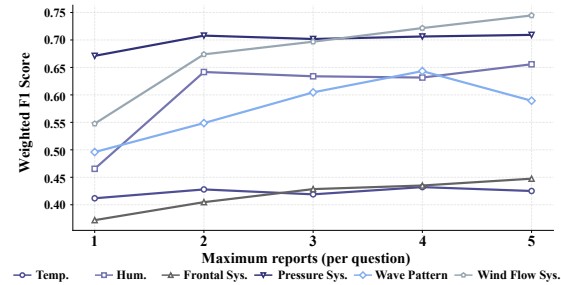

*Figure 4.* Performances with increasing number of reports for each question

generalization. As shown in Figure 5(B) and (C), our model consistently outperforms closed-source models in these cross-region evaluations, demonstrating strong generalization across distant regions. The consistent gains across multiple weather aspects indicate that the model captures underlying meteorological dynamics rather than overfitting to region-specific surface patterns.

### 4.7. City-level Analysis

We further analyze model performance at the city level, as shown in Figure 6. Each radar chart displays performance across all meteorological dimensions for a representative city from distinct geographic regions. **WeatherSyn** outperforms all baselines in each city. We observe pronounced geographic heterogeneity. The performance gap between **WeatherSyn** and the baselines is more prominent in Honolulu and Flagstaff. This disparity can likely be attributed

to the complexity of their local environments: Honolulu features maritime and island-driven weather dynamics, while Flagstaff is influenced by high-altitude plateau terrain. In contrast, Charleston exhibits more uniform performance across models, suggesting that its weather patterns are comparatively simpler and more easily captured by existing approaches. Beyond superior average performance, these results demonstrate that our model offers exceptional stability and adaptability across individual cities, which is critical for real-world weather report generation.

## 5. Related Works

**Domain-Specific Report Generation** Multimodal large language models (MLLMs) are increasingly applied to automated report generation in domain-specific tasks that demand structured and context-aware summaries. In radiology, for instance, MLLMs generate diagnostic reports directly from medical images such as X-rays and CT scans (Li et al., 2023; Moor et al., 2023). Similarly, remote sensing leverages MLLMs to produce textual descriptions from satellite and aerial imagery (Pang et al., 2025; Kuckreja et al., 2024; Zhang et al., 2024; Chen et al., 2025), supporting applications including land use monitoring, disaster assessment, and environmental analysis. Despite these advancements, the use of MLLMs for weather report generation which requires interpreting meteorological signals to produce public-facing forecasts remains underexplored.

**Earth VLMs** To capture the alignment between variables and textual descriptions, WeatherQA (Ma et al., 2024) further enables summary generation but remains centered on

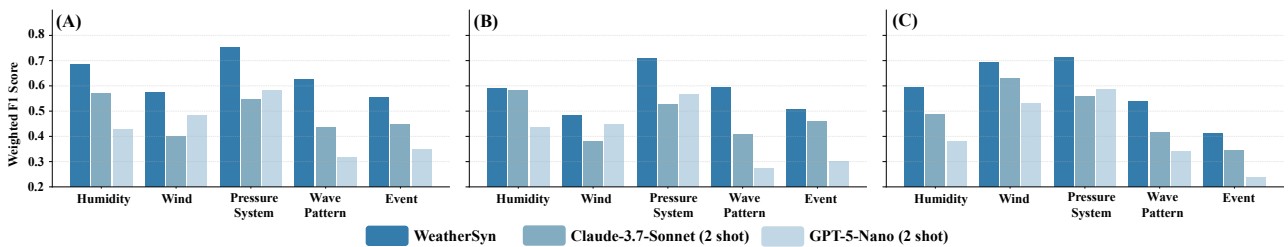

*Figure 5.* (A) Trained on randomly selected cities, tested on other cities. (B) Trained on cities in the northern United States, tested on cities in the southern United States. (C) Trained on cities in the eastern United States, tested on cities in the western United States.

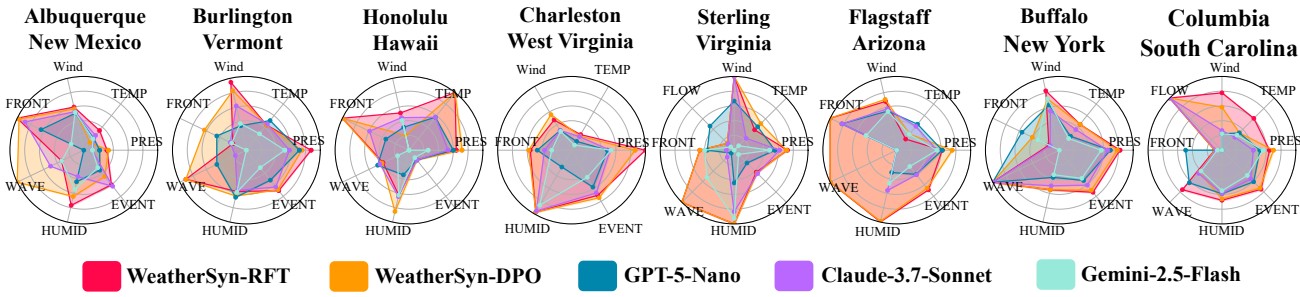

*Figure 6.* Regional performance evaluation based on weighted F1 Score across representative U.S. cities.

severe weather event prediction. Similarly, CLLMate (Li et al., 2024) proposed a multimodal dataset for reasoning about weather parameters and predicting severe weather in real-world scenarios, formulating the problem as a multiple-choice task through a hierarchical categorization of weather and climate events. OmniEarth-Bench (Wang et al., 2025) introduced a multimodal Earth science dataset to benchmark existing MLLMs across multiple dimensions. However, it primarily focuses on Earth science knowledge questions in a multiple-choice format. These works exhibit two notable limitations: (1) their multiple-choice design, while suitable for classification-style evaluation, is inherently limited in assessing free-form, context-rich weather report generation, and thus falls short of supporting general forecasting report; and (2) their focus is largely confined to severe weather events, leaving the broader scope of everyday, fine-grained weather reporting underexplored.

## 6. Conclusion

In this work, we propose the WFR task, which aims to generate comprehensive textual weather prediction reports based on initial weather conditions. To address data scarcity in this domain, we construct **WSInstruct**, the first instruction-tuning multimodal dataset encompassing both weather conditions and textual reports across 31 cities. We further fine-tune the open-source MLLM Qwen3-VL-8B, resulting in **WeatherSyn**. Our model not only surpasses closed-source MLLMs in performance but also exhibits strong cross-city transferability. Experimental results show that **WeatherSyn** captures distinct climate-dependent weather patterns across cities and demonstrates robust performance in long-range forecasting. In future work, we plan to incorporate data from additional cities and train a global report model with stronger generalization capability.

## Impact Statement

Climate change has intensified weather variability and increased the frequency of extreme events such as heatwaves, droughts, and heavy rainfall, leading to substantial socioeconomic losses. Accurate and comprehensive weather forecast reports are essential for enabling individuals, communities, and policymakers to make informed decisions in daily life, agriculture, and transportation systems.

While several specialized weather-focused multimodal large language models have been proposed, existing approaches primarily target simplified tasks such as multiple-choice question answering or the generation of severe weather alerts. In practice, weather forecasting reports more closely resemble urban-scale decision-support documents rather than narrowly defined classification or alert-generation tasks. Improving models' ability to generate accurate and coherent general-purpose weather forecast reports, therefore has the potential to enhance public preparedness, operational planning, and resilience to climate-related risks.

## Acknowledgements

This work is supported by NSFC 62572418 and Guangdong Provincial Talent Program(No.2024TQ08X366). Yang Liu is supported in part by the Postdoctoral Fellowship Scheme of The Chinese University of Hong Kong.

## Limitation

Due to the limited availability of open-source weather forecast report data, our experiments primarily focus on cities in the United States. In future work, we plan to collect forecast reports from a broader range of countries and develop models that generalize more effectively across diverse geographic regions. Additionally, our current approach only leverages the initial weather state, as our goal is to analyze the model's forecasting and reasoning capabilities under minimal input conditions. Extending the framework to incorporate multiple prediction steps from HRES as visual inputs could provide richer contextual information and further enhance performance. The corresponding results are presented in Appendix E.1.

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

*Table 5.* Counts of Claims by Aspect and Category Across Regions

| Region | Temperature | | | Wind | | Humidity | | Frontal System | | Pressure System | | Wave Pattern | | Wind Flow System | | Event | | |
|---|---|---|---|---|---|---|---|---|---|---|---|---|---|---|---|---|---|---|
| | Hot | Cool | Moderate | Strong | Light | Dry | Moist | Cold Fr. | Warm Fr. | High Pr. | Low Pr. | Ridge | Trough | Onshore | Offshore | Precipitation | Snow | Storm |
| Albany, New York | 74 | 209 | 142 | 78 | 45 | 132 | 59 | 100 | 46 | 119 | 103 | 6 | 7 | 0 | 0 | 332 | 151 | 169 |
| Albuquerque, New Mexico | 218 | 208 | 55 | 130 | 99 | 161 | 70 | 135 | 2 | 86 | 45 | 10 | 42 | 0 | 0 | 215 | 100 | 231 |
| Binghamton, New York | 87 | 116 | 78 | 45 | 14 | 120 | 51 | 94 | 54 | 128 | 59 | 9 | 10 | 0 | 0 | 283 | 197 | 185 |
| Blacksburg, Virginia | 49 | 129 | 18 | 60 | 13 | 90 | 47 | 213 | 37 | 264 | 172 | 9 | 17 | 5 | 3 | 212 | 67 | 120 |
| Boston, Massachusetts | 92 | 196 | 169 | 89 | 42 | 239 | 53 | 130 | 59 | 170 | 121 | 2 | 4 | 9 | 7 | 300 | 100 | 129 |
| Buffalo, New York | 97 | 161 | 39 | 67 | 23 | 131 | 33 | 116 | 56 | 188 | 140 | 7 | 19 | 0 | 0 | 298 | 237 | 133 |
| Burlington, Vermont | 68 | 139 | 107 | 57 | 39 | 149 | 29 | 70 | 39 | 144 | 118 | 6 | 27 | 0 | 0 | 331 | 203 | 94 |
| Caribou, Maine | 33 | 28 | 4 | 3 | 0 | 20 | 26 | 158 | 50 | 284 | 195 | 12 | 23 | 0 | 1 | 27 | 5 | 17 |
| Charleston, West Virginia | 52 | 71 | 10 | 14 | 3 | 61 | 18 | 191 | 37 | 193 | 74 | 5 | 13 | 1 | 3 | 155 | 78 | 74 |
| Cleveland, Ohio | 18 | 29 | 6 | 4 | 5 | 16 | 5 | 199 | 74 | 267 | 221 | 61 | 63 | 0 | 0 | 26 | 22 | 33 |
| Columbia, South Carolina | 58 | 106 | 48 | 17 | 38 | 194 | 82 | 222 | 54 | 188 | 114 | 60 | 30 | 17 | 1 | 213 | 4 | 125 |
| Flagstaff, Arizona | 146 | 166 | 31 | 142 | 128 | 170 | 35 | 59 | 0 | 21 | 43 | 0 | 5 | 0 | 0 | 244 | 104 | 134 |
| Great Falls, Montana | 156 | 209 | 100 | 135 | 113 | 143 | 6 | 89 | 3 | 80 | 29 | 5 | 7 | 0 | 0 | 243 | 217 | 142 |
| Greer, South Carolina | 92 | 146 | 29 | 12 | 12 | 196 | 139 | 239 | 30 | 255 | 117 | 6 | 20 | 0 | 2 | 180 | 31 | 135 |
| Honolulu, Hawaii | 9 | 57 | 4 | 135 | 341 | 110 | 86 | 133 | 8 | 87 | 77 | 18 | 58 | 0 | 0 | 441 | 0 | 84 |
| Las Vegas, Nevada | 90 | 136 | 45 | 124 | 86 | 98 | 34 | 18 | 0 | 23 | 51 | 1 | 7 | 0 | 0 | 193 | 73 | 115 |
| Los Angeles, California | 229 | 224 | 47 | 136 | 53 | 121 | 45 | 39 | 0 | 74 | 67 | 10 | 19 | 42 | 69 | 249 | 66 | 99 |
| Mt. Holly, New Jersey | 8 | 16 | 0 | 15 | 5 | 17 | 6 | 208 | 45 | 223 | 189 | 23 | 36 | 2 | 19 | 16 | 6 | 29 |
| Newport, North Carolina | 4 | 24 | 0 | 8 | 0 | 32 | 0 | 183 | 23 | 243 | 91 | 13 | 16 | 0 | 9 | 8 | 1 | 30 |
| Portland, Maine | 88 | 154 | 44 | 59 | 38 | 77 | 41 | 154 | 41 | 249 | 234 | 29 | 31 | 16 | 5 | 219 | 86 | 92 |
| Portland, Oregon | 52 | 132 | 50 | 64 | 31 | 82 | 42 | 84 | 40 | 56 | 60 | 47 | 51 | 58 | 45 | 316 | 118 | 52 |
| Raleigh/Durham, North Carolina | 47 | 131 | 21 | 18 | 8 | 64 | 37 | 218 | 45 | 296 | 99 | 10 | 44 | 2 | 11 | 90 | 0 | 83 |
| San Diego, California | 178 | 223 | 50 | 171 | 69 | 134 | 68 | 10 | 0 | 101 | 90 | 11 | 38 | 76 | 82 | 211 | 50 | 73 |
| San Francisco Bay Area, California | 180 | 241 | 84 | 82 | 57 | 183 | 19 | 41 | 4 | 73 | 28 | 13 | 32 | 61 | 52 | 203 | 14 | 35 |
| Seattle, Washington | 75 | 110 | 16 | 63 | 23 | 113 | 12 | 131 | 39 | 63 | 68 | 91 | 100 | 44 | 23 | 315 | 77 | 42 |
| Spokane, Washington | 72 | 152 | 58 | 107 | 82 | 134 | 11 | 69 | 3 | 26 | 23 | 4 | 2 | 0 | 0 | 293 | 220 | 111 |
| State College, Pennsylvania | 62 | 171 | 73 | 67 | 38 | 113 | 68 | 163 | 41 | 132 | 91 | 22 | 26 | 0 | 0 | 219 | 124 | 109 |
| Sterling, Virginia | 15 | 24 | 6 | 3 | 2 | 14 | 5 | 206 | 68 | 242 | 151 | 10 | 9 | 0 | 17 | 28 | 5 | 32 |
| Wakefield, Virginia | 25 | 50 | 9 | 10 | 9 | 58 | 8 | 231 | 44 | 305 | 182 | 4 | 17 | 1 | 8 | 45 | 4 | 33 |
| Wilmington, North Carolina | 110 | 207 | 68 | 41 | 46 | 169 | 47 | 233 | 29 | 225 | 99 | 8 | 18 | 6 | 11 | 257 | 7 | 114 |
| Wilmington, Ohio | 88 | 135 | 48 | 49 | 21 | 204 | 63 | 174 | 31 | 262 | 169 | 17 | 28 | 0 | 0 | 253 | 69 | 131 |

# A. More Dataset Details

## A.1. Details of Textual Data

We perform weather forecast report in 31 cities, including Albuquerque (New Mexico), Wakefield (Virginia), Albany (New York), Binghamton (New York), Boston (Massachusetts), Burlington (Vermont), Buffalo (New York), Columbia (South Carolina), Caribou (Maine), Cleveland (Ohio), State College (Pennsylvania), Flagstaff (Arizona), Greer (South Carolina), Portland (Maine), Honolulu (Hawaii), Wilmington (North Carolina), Wilmington (Ohio), Los Angeles (California), Sterling (Virginia), Newport (North Carolina), San Francisco Bay Area (California), Spokane (Washington), Mt. Holly (New Jersey), Portland (Oregon), Raleigh/Durham (North Carolina), Charleston (West Virginia), Blacksburg (Virginia), Seattle (Washington), San Diego (California), Great Falls (Montana), Las Vegas (Nevada). We retrieve the synopsis data from the Area Forecast Discussion text products provided by the Iowa Environmental Mesonet[1], which archive past short- and medium-range weather forecast synopsis.

## A.2. Dataset Statistics

We focus on 8 weather aspects and 18 weather categories. The number of extracted claims are shown in Table 5. The count of aspects is shown in Table 7. We observe substantial variations in claim distributions across regions, which can be attributed to differences in local climatic conditions and reporting styles.

Specifically, regions located in colder or northern areas (e.g., Burlington, Vermont; Caribou, Maine; Spokane, Washington) exhibit a higher frequency of claims related to snow and cold frontal systems, whereas southern and coastal regions (e.g., Honolulu, Hawaii; Los Angeles, California; San Diego, California) contain more claims describing precipitation, moderate temperatures, and onshore wind flow patterns. Inland or arid regions, such as Flagstaff, Arizona and Las Vegas, Nevada, show relatively more claims associated with temperature extremes and dry humidity conditions.

In addition to climatic factors, the description style of weather reports also contributes to these discrepancies. Some regions tend to provide more detailed narratives, resulting in a higher number of fine-grained claims across multiple categories, while others adopt more concise reporting styles that emphasize only dominant weather phenomena. This regional imbalance highlights the inherent heterogeneity of weather report data.

---

[1] https://mesonet.agron.iastate.edu/wx/afos/

*Table 6.* Data Statistics by city (SFT, RFT, DPO, Test)

| Region | SFT | RFT | DPO | Test | Region | SFT | RFT | DPO | Test |
|---|---|---|---|---|---|---|---|---|---|
| Albany, New York | 230 | 728 | 48 | 49 | Los Angeles, California | 230 | 724 | 49 | 42 |
| Albuquerque, New Mexico | 216 | 684 | 45 | 46 | Mt. Holly, New Jersey | 164 | 540 | 29 | 33 |
| Binghamton, New York | 213 | 684 | 42 | 43 | Newport, North Carolina | 138 | 464 | 22 | 34 |
| Blacksburg, Virginia | 231 | 744 | 45 | 45 | Portland, Maine | 226 | 704 | 50 | 43 |
| Boston, Massachusetts | 217 | 704 | 41 | 48 | Portland, Oregon | 174 | 592 | 26 | 34 |
| Buffalo, New York | 227 | 728 | 45 | 42 | Raleigh, North Carolina | 200 | 636 | 41 | 41 |
| Burlington, Vermont | 217 | 688 | 45 | 43 | San Diego, California | 201 | 652 | 38 | 48 |
| Caribou, Maine | 182 | 576 | 38 | 47 | San Francisco Bay Area, California | 196 | 636 | 37 | 43 |
| Charleston, West Virginia | 199 | 644 | 38 | 40 | Seattle, Washington | 201 | 664 | 35 | 43 |
| Cleveland, Ohio | 193 | 612 | 40 | 38 | Spokane, Washington | 200 | 620 | 45 | 34 |
| Columbia, South Carolina | 216 | 716 | 37 | 33 | State College, Pennsylvania | 212 | 708 | 35 | 48 |
| Flagstaff, Arizona | 201 | 644 | 40 | 48 | Sterling, Virginia | 162 | 536 | 28 | 35 |
| Great Falls, Montana | 227 | 720 | 47 | 45 | Wakefield, Virginia | 192 | 596 | 43 | 40 |
| Greer, South Carolina | 233 | 740 | 48 | 47 | Wilmington, North Carolina | 234 | 764 | 43 | 41 |
| Honolulu, Hawaii | 205 | 672 | 37 | 40 | Wilmington, Ohio | 241 | 780 | 46 | 42 |
| Las Vegas, Nevada | 166 | 512 | 38 | 37 | | | | | |

*Table 7.* Count of aspects in training and test set.

| Count of Aspect | Temperature | Wind | Humidity | Frontal System | Pressure System | Wave Pattern | Wind Flow System | Event |
|---|---|---|---|---|---|---|---|---|
| Training Set | 16262 | 6976 | 9580 | 10624 | 16774 | 2696 | 1416 | 23672 |
| Test Set | 1882 | 740 | 1008 | 956 | 1388 | 249 | 112 | 1753 |

## A.3. Alignment of the ERA5 and Report

We first convert the UTC timestamps in the ERA5 data to the local time for each city. Since the temporal resolution is 6 hours, we select the report issued within the following 3 hours as the corresponding report for the ERA5 visual input. As multiple reports may be issued throughout a single day, we use the one released in the early morning as the representative report for that day.

## A.4. Details of WSInstruct-Pressure and WSInstruct-Global

**WSInstruct-Pressure**  To explore the impact of incorporating pressure-level variables, we construct an extended dataset, called **WSInstruct-Pressure**. Specifically, we supplement the regional single-level data with pressure-level variables including geopotential, specific humidity, temperature, and the u- and v-components of wind at four standard pressure levels: 200 hPa, 500 hPa, 700 hPa, and 850 hPa. This enriched dataset allows us to assess whether vertical atmospheric structure contributes to improved model performance.

**WSInstruct-Global**  To evaluate the influence of broader spatial context, we create another dataset, **WSInstruct-Global**, by incorporating large-scale geographical information. We add single-level data covering the entire CONUS area. This setup allows us to examine how the incorporation of a wider regional view affects the forecasting ability of the model.

## A.5. Geographic Region Split and Mapping

To evaluate the model's generalization capability across different geographic regions, we partition the dataset into four distinct regions based on geographic coordinates: Northeast (NE), Northwest (NW), Southeast (SE), and Southwest (SW). The regional division is defined using latitude and longitude thresholds. Specifically, cities with latitude $\geq 40°$ are classified as North (Northeast and Northwest), while those with latitude $< 40°$ are classified as South (Southeast and Southwest). Similarly, cities with longitude $\geq -100°$ are classified as East (Northeast and Southeast), whereas those with longitude $< -100°$ are classified as West (Northwest and Southwest).

This partitioning strategy facilitates a systematic evaluation of the model's performance across geographically and climatically diverse regions. Table 8 provides the complete mapping from city abbreviations to their corresponding full city names

| Region | Code | City Name |
|--------|------|-----------|
| Northeast | ALY | Albany, New York |
| | BGM | Binghamton, New York |
| | BOX | Boston, Massachusetts |
| | BTV | Burlington, Vermont |
| | BUF | Buffalo, New York |
| | CAR | Caribou, Maine |
| | CLE | Cleveland, Ohio |
| | CTP | State College, Pennsylvania |
| | GYX | Portland, Maine |
| Northwest | OTX | Spokane, Washington |
| | PQR | Portland, Oregon |
| | SEW | Seattle, Washington |
| | TFX | Great Falls, Montana |
| Southeast | AKQ | Wakefield, Virginia |
| | CAE | Columbia, South Carolina |
| | GSP | Greer, South Carolina |
| | ILM | Wilmington, North Carolina |
| | ILN | Wilmington, Ohio |
| | LWX | Sterling, Virginia |
| | MHX | Newport, North Carolina |
| | PHI | Mt. Holly, New Jersey |
| | RAH | Raleigh, North Carolina |
| | RLX | Charleston, West Virginia |
| | RNK | Blacksburg, Virginia |
| Southwest | ABQ | Albuquerque, New Mexico |
| | FGZ | Flagstaff, Arizona |
| | HFO | Honolulu, Hawaii |
| | LOX | Los Angeles, California |
| | MTR | San Francisco Bay Area, California |
| | SGX | San Diego, California |
| | VEF | Las Vegas, Nevada |

*Table 8.* City abbreviations and corresponding full names by region

for each of the four regions.

For the geographically close generalization experiment, a subset of cities is randomly selected as training cities, while the remaining cities within the same regions are used for testing. The abbreviations of the training and test cities are listed below:

**Training cities:**
VEF, SGX, RAH, BGM, TFX, BOX, CAE, GYX, PQR, ILM, ILN, CLE, MHX, OTX, CAR

**Test cities:**
ABQ, AKQ, ALY, BTV, BUF, CTP, FGZ, GSP, HFO, LOX, LWX, MTR, PHI, RLX, RNK, SEW

## B. Additional Related Work

**Weather Prediction**    Numerical Weather Prediction (NWP) relies on the equations of thermodynamics and fluid dynamics to model the dynamic interactions among the atmosphere, land, and ocean systems. With the advancement of deep learning, numerous studies have explored the use of radar imagery (Veillette et al., 2020) for precipitation nowcasting (Wen et al., 2024; Gao et al., 2023), while Digital Typhoon (Kitamoto et al., 2023) leverages typhoon satellite imagery for spatiotemporal forecasting. However, these works often focus on a single meteorological variable and lack accompanying textual event narratives. The release of the ECMWF Reanalysis v5 (ERA5) dataset (Hersbach et al., 2020) has further enabled the

*Table 9.* The definition of single level variable.

| Variable | Definition |
| --- | --- |
| land sea mask | The proportion of land, as opposed to ocean or inland waters |
| 10m u component of wind | The eastward component of the 10m wind. It is the horizontal speed of air moving towards the east, at a height of ten metres above the surface of the Earth. |
| 10m v component of wind | The northward component of the 10m wind. It is the horizontal speed of air moving towards the north, at a height of ten metres above the surface of the Earth. |
| 2m temperature | The temperature of air at 2m above the surface of land, sea or in-land waters. |
| mean sea level pressure | The pressure of the atmosphere adjusted to the height of mean sea level. |
| sea surface temperature | The temperature of sea water near the surface. |
| snow depth | The depth of snow from the snow-covered area. |
| surface pressure | The pressure of the atmosphere on the surface of land, sea and in-land water. |
| total cloud cover | The proportion of a grid box area covered by cloud. |
| total precipitation 6hr | The total precipitation over the past 6 hours. |
| total column water vapour | The total amount of water vapour in a vertical column of the atmosphere, from the surface to the top of the atmosphere. |
| total column water | The total amount of liquid water in a vertical column of the atmosphere. |

development of weather foundation models such as FourcastNet (Pathak et al., 2022), GraphCast (Lam et al., 2022), FuXi (Chen et al., 2023a), and Pangu-Weather (Bi et al., 2023). Nevertheless, these models primarily focus on predicting numerical meteorological variables rather than directly producing textual forecasts for public communication.

## C. Evaluation

### C.1. Consistency Check

To evaluate extraction, we treat a claim at any forecasting time step as a true positive (TP) if its claim category matches the ground truth. Claims that appear in the ground truth but are missing in the extraction are counted as false negatives (FN), while extra claims beyond the reference are counted as false positives (FP).

We aggregate the numbers of TP, FN, and FP across all forecasting time steps and all samples. Based on these globally accumulated counts, we compute precision, recall, and F1-score as:

$$\text{Precision} = \frac{\sum TP}{\sum TP + \sum FP}, \quad \text{Recall} = \frac{\sum TP}{\sum TP + \sum FN}, \tag{3}$$

$$\text{F1} = \frac{2 \cdot \text{Precision} \cdot \text{Recall}}{\text{Precision} + \text{Recall}}. \tag{4}$$

This global aggregation strategy jointly considers all forecasting horizons and samples, and naturally penalizes both missing claims (FN) and over-extracted claims (FP), yielding a single extraction F1 score that reflects the overall extraction performance of the model.

### C.2. Test Set Evaluation

To provide a fair and balanced evaluation across claim categories, we compute claim- and aspect-level weighted metrics. Specifically, for each claim $c$ belonging to aspect $a$, let $TP_{a,c}$, $FP_{a,c}$, and $FN_{a,c}$ denote the numbers of true positives, false

positives, and false negatives, respectively. The claim-level precision and recall are computed as

$$\text{Precision}_{a,c} = \frac{\text{TP}_{a,c}}{\text{TP}_{a,c} + \text{FP}_{a,c}}, \quad \text{Recall}_{a,c} = \frac{\text{TP}_{a,c}}{\text{TP}_{a,c} + \text{FN}_{a,c}}.$$

For each aspect $a$, we assign a weight to each claim $c \in a$ inversely proportional to its frequency, $w_{a,c} = 1/(\text{TP}_{a,c} + \text{FN}_{a,c})$, and normalize the weights as

$$\tilde{w}_{a,c} = \frac{w_{a,c}}{\sum_{c \in a} w_{a,c}}.$$

The aspect-level weighted precision, recall, and F1-score are then computed as

$$\text{Weighted Precision}_a = \sum_{c \in a} \tilde{w}_{a,c} \, \text{Precision}_{a,c}, \quad \text{Weighted Recall}_a = \sum_{c \in a} \tilde{w}_{a,c} \, \text{Recall}_{a,c},$$

$$\text{Weighted F1}_a = \frac{2 \cdot \text{Weighted Precision}_a \cdot \text{Weighted Recall}_a}{\text{Weighted Precision}_a + \text{Weighted Recall}_a}.$$

## D. Implementation Details

We employ the open-source Qwen3-VL-8B as our backbone model, and our implementation is built on DeepSpeed, HuggingFace, and the LLaMA Factory library. In both the SFT and RFT stages, we set the learning rates for the LLM, merger, and vision encoder to 1e-5, 1e-5, and 2e-6, respectively. The model is trained for one epoch with a per-GPU batch size of 2 on 8 NVIDIA A800 GPUs. During the DPO stage, the learning rate is reduced to 5e-7, the vision tower is frozen, and the per-GPU batch size is set to 1, with the $\beta$ hyperparameter fixed at 0.1.

For sampling during both the RFT and DPO stages, we generate 40 candidate reports per input using a temperature of 0.9, top-k of 50, and top-p of 0.9 to encourage output diversity.

During inference, the temperature is set to 0.3 for all models, and the maximum number of generated tokens is limited to 400. For evaluation, we locally deploy Qwen-2.5-72B using vLLM, with the maximum model length set to 4096.

## E. Additional Experiments

### E.1. NWP-augmented Training

Weather forecasting pipelines are not limited to using only the initial state as visual input. In fact, NWP forecasting results can also be incorporated to provide additional visual information. Specifically, we retrieve the HRES forecasting outputs from WeatherBench2, which include the 00 and 12 UTC initializations of HRES along with their corresponding 6-hourly forecasts. We select six single-level variables: 10 m u-component of wind, 10 m v-component of wind, 2 m temperature, mean sea level pressure, surface pressure, and 6-hour total precipitation. To reduce computational complexity caused by the large number of images, we compute the mean value of each prediction over 24 hours, thereby obtaining daily prediction variables spanning from 1 to 4 days. During training, we replace the visual data in $\mathcal{D}_{RFT}$ with variables from different prediction horizons, ranging from only 1-day predictions to the full 1–4-day forecasts, in order to assess whether incorporating longer-range forecast information can further enhance performance. The results are reported in Figure 7. We can observe that as the time horizon of the visual input increases, the model's performance improves accordingly, demonstrating that incorporating NWP prediction results can further enhance performance. A marginal effect is observed as the time horizon extends from 2 days to 3 days.

### E.2. Further Result on Weather Aspect Analysis

The precision results are shown in Table 10. The **WeatherSyn** consistently outperforms all baseline models on the **WSInstruct** test set in terms of precision. Among closed-source MLLMs, Claude-3.7-Sonnet and Gemini-2.5-flash achieve relatively better performance on structurally explicit aspects such as Wave Pattern and Wind Flow System. However, their overall precision remains limited, with average scores ranging from 0.49 to 0.53, indicating that general-purpose MLLMs struggle with fine-grained and domain-specific weather information extraction.

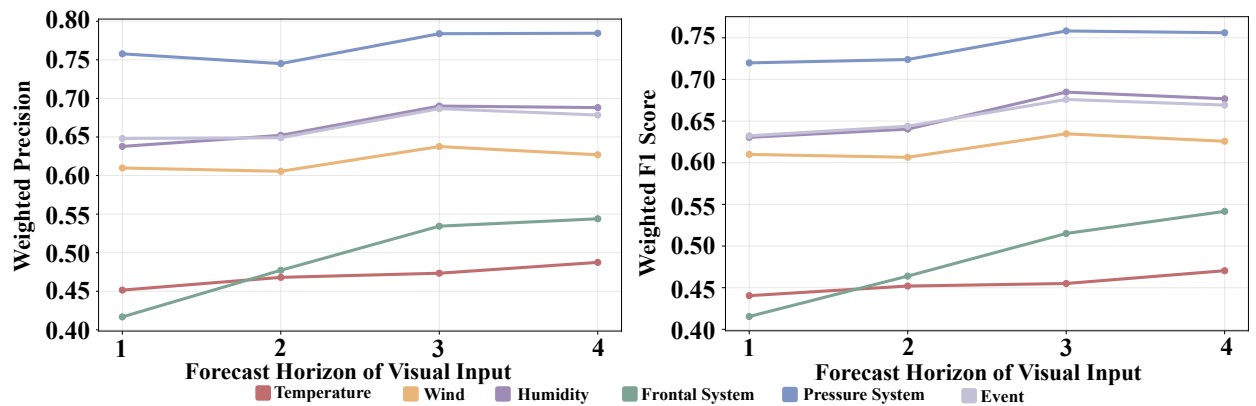

*Figure 7.* The impact of different forecast horizon of the visual input on training

*Table 10.* Precision result of **WeatherSyn** and other baselines on **WSInstruct** test set.

| Algorithms | Temperature | Wind | Humidity | Frontal System | Pressure System | Wave Pattern | Wind Flow System | Event | **Average** |
|---|---|---|---|---|---|---|---|---|---|
| *Closed-source MLLMs* | | | | | *Precision* | | | | |
| GPT-4.1-mini (2-shot) | 0.35 | 0.52 | 0.37 | 0.29 | 0.62 | 0.56 | 0.53 | 0.71 | 0.49 |
| GPT-5-nano (2-shot) | 0.36 | 0.53 | 0.37 | 0.32 | 0.63 | 0.48 | 0.53 | 0.78 | 0.50 |
| Gemini-2.5-flash (zero-shot)* | 0.35 | 0.55 | 0.44 | 0.33 | 0.59 | 0.52 | 0.60 | 0.76 | 0.51 |
| Claude-3.7-Sonnet (2-shot) | 0.39 | 0.55 | 0.45 | 0.31 | 0.58 | 0.66 | 0.67 | 0.69 | 0.53 |
| *Open-source MLLMs* | | | | | | | | | |
| WeatherQA | 0.34 | 0.55 | 0.49 | 0.47 | 0.60 | 0.47 | 0.49 | 0.64 | 0.50 |
| OmniEarth | 0.35 | 0.57 | 0.42 | 0.35 | 0.72 | 0.56 | 0.52 | 0.63 | 0.51 |
| **WeatherSyn** | 0.42 | 0.62 | 0.63 | 0.43 | 0.75 | 0.64 | 0.67 | 0.66 | 0.60 |
| **WeatherSyn-RFT** | 0.44 | 0.62 | 0.69 | 0.47 | 0.75 | 0.64 | 0.74 | 0.66 | 0.62 |
| **WeatherSyn-DPO** | 0.44 | 0.63 | 0.67 | 0.49 | 0.77 | 0.64 | 0.72 | 0.67 | 0.62 |

In contrast, WeatherSyn achieves substantial and consistent improvements across all eight weather aspects, reaching an average precision of 0.60 and surpassing all closed-source and open-source baselines. This demonstrates the effectiveness of domain-specific instruction tuning for weather forecasting report generation.

### E.3. Data Volume Analysis

We examine the impact of data volume and diversity on model performance by varying the maximum number of reports retained per case during RFT. The precision results are shown in Figure 8, where performance consistently improves as the training corpus expands. Specifically, increasing the number of reports per case from 1 to 4 leads to gains of 0.05 and 0.18 in the *Temperature* and *Wind Flow System* aspects, respectively. These results suggest that greater lexical diversity in the training data enhances the model's ability to align meteorological visual inputs with their corresponding textual descriptions.

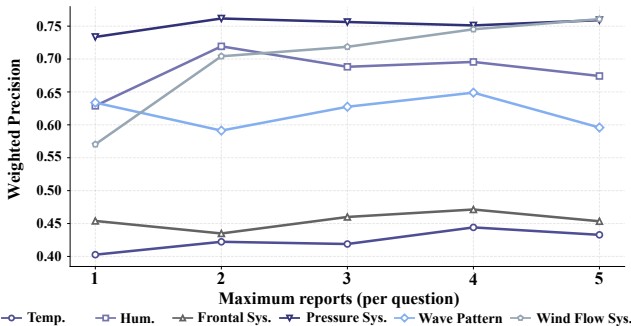

*Figure 8.* Performances with increasing number of reports for each question

We further report the reference-based metric with the results summarized in Table 11. Overall, increasing the number of reports leads to consistent improvements across reference-based metrics, including BLEU-1, ROUGE-L, and METEOR, indicating that a larger training corpus benefits report generation quality. Performance peaks when up to three reports per

case are used, yielding the highest scores on all three metrics, while marginal degradation is observed as more reports are added. This suggests that moderate data expansion enhances lexical diversity and semantic alignment, whereas excessive reports may introduce redundancy or noise, limiting further gains.

*Table 11.* Effect of Training Data Volume on Reference-based Evaluation Metrics

| Maximum reports | 1 | 2 | 3 | 4 | 5 |
|---|---|---|---|---|---|
| BLEU-1 | 0.4369 | 0.4394 | **0.4426** | 0.4367 | 0.4276 |
| ROUGE-L | 0.3120 | 0.3171 | **0.3175** | 0.3152 | 0.3081 |
| METEOR | 0.2477 | 0.2513 | **0.2525** | 0.2508 | 0.2471 |

### E.4. Visual Input Analysis

To investigate how different types of meteorological visual inputs affect model performance, we construct two extended datasets based on the default regional single-level **WSInstruct**.

First, to capture multi-level atmospheric information beyond single-level variables, we introduce **WSInstruct-Pressure**, which augments the base data with pressure-level variables. Second, to assess the effect of the broader geographical context, we build **WSInstruct-Global** by incorporating larger-scale regional data. The detailed configuration is shown in Appendix A.4. We fine-tune the model under each visual input configuration during RFT and report the F1 Score results on their respective test sets in Table 12. Overall, we observe that incorporating additional pressure-level variables does not lead to consistent performance improvements over the base single-level setting.

Similarly, expanding the spatial context by introducing larger-scale regional information yields only limited benefits. Although broader geographical inputs can provide large-scale weather systems, they may also introduce redundant or noisy information that is less directly aligned with city-level weather report generation. As a result, the model shows comparable or slightly degraded performance on several aspects.

*Table 12.* Performance comparison under different meteorological visual input configurations on the WSInstruct test set.

| Dataset | Temperature | Wind | Humidity | Frontal System | Pressure System | Wave Pattern | Wind Flow System | Event | **Average** |
|---|---|---|---|---|---|---|---|---|---|
| **WSInstruct-Pressure** | 0.44 | 0.61 | 0.61 | 0.43 | 0.72 | 0.62 | 0.70 | 0.63 | 0.59 |
| **WSInstruct-Global** | 0.42 | 0.59 | 0.65 | 0.41 | 0.70 | 0.61 | 0.61 | 0.63 | 0.57 |
| **WSInstruct** | 0.43 | 0.62 | 0.63 | 0.43 | 0.70 | 0.64 | 0.72 | 0.66 | 0.59 |

### E.5. Generalization Analysis

Figure 9 reports the weighted precision scores for the same cross-region splits described in Section 4.6. Consistent with the F1-score trends in the main text, our model achieves higher precision than Claude-3-7-Sonnet and GPT-5-Nano across most weather aspects and generalization settings. Notably, the gains are more pronounced for structurally driven aspects such as *Pressure System* and *Wave Pattern*, indicating that the model produces more accurate and less spurious predictions under both geographically proximate and distant transfer scenarios. Although precision varies across aspects due to differences in regional climate complexity, the overall consistency across (A)–(C) further confirms that the observed generalization improvements are not driven by recall alone, but reflect more precise alignment between generated reports and underlying meteorological conditions.

## F. Prompt

### F.1. Temporal Forecast Extraction

The Instruction prompt for extracting report for different time step and an example is provided as follows:

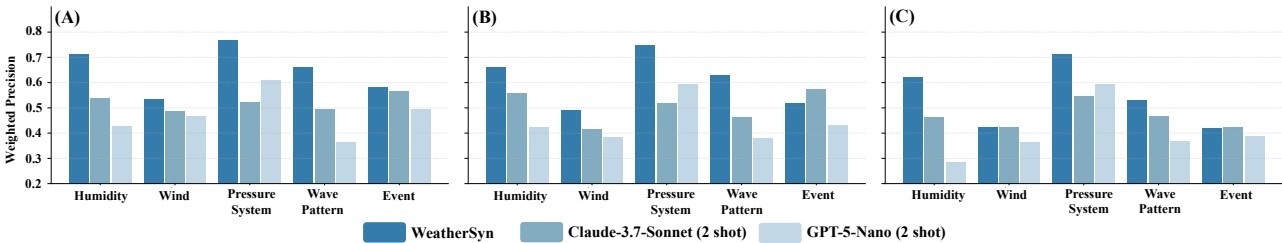

*Figure 9.* (A) Trained on randomly selected cities, tested on other cities. (B) Trained on cities in the northern United States, tested on cities in the southern United States. (C) Trained on cities in the eastern United States, tested on cities in the western United States.

### Temporal Forecast Extraction Prompt

```
You are an expert meteorological forecaster. Your task is to extract structured daily forecast information
↪   from the user's text.

### STRICT RULES

- Use ONLY the factual content provided in the original forecast.
- DO NOT introduce new facts.
- DO NOT guess missing information.
- For any day where no explicit forecast is present, return an empty string.
- Output MUST be valid JSON with the EXACT structure shown below.
- Do NOT include commentary, explanation, or any text outside the JSON.
- When splitting or extracting forecast content, **each output must be a complete, grammatically correct
↪   sentence.**
- **Do NOT drop the subject or any essential clause** from the original sentence.
- **Do NOT output sentence fragments. Every extracted entry must retain the original meaning and sentence
↪   structure.**
- **If one sentence contains multiple time-specific clauses, you must split them into separate complete
↪   sentences, each with a proper subject.**

### EXAMPLE OF REQUIRED SPLITTING

Original: A cold front approaches the region from the west on Wednesday and likely stalls or washes out over
↪   the area into Thursday.
Correct split:
1. A cold front approaches the region from the west on Wednesday.
2. The cold front washes out over the area into Thursday.

### REQUIRED JSON OUTPUT FORMAT

<json_format_template>

Extract the corresponding forecast content for each date from the text below.
ORIGINAL FORECAST TEXT:

<original_forecast_test>

Return ONLY the JSON. No explanations.
```

**Temporal Forecast Extraction Example**

```
<json_format_template>:

"daily_forecast": [
    {
        "date": "20190927",
        "weekday": "Friday",
        "forecast": ""
    },
    {
        "date": "20190928",
        "weekday": "Saturday",
        "forecast": ""
    },
    {
        "date": "20190929",
        "weekday": "Sunday",
        "forecast": ""
    },
    {
        "date": "20190930",
        "weekday": "Monday",
        "forecast": ""
    }

<original_forecast_test>:

A high pressure ridge over New England will bring mainly fair and cool weather tonight. On Saturday, a warm
↪  front followed by a cold front will bring a threat of showers and thunderstorms. On Sunday, high pressure
↪  building in from Ontario will bring fair and cool weather. Monday will start out fair, but an approaching
↪  warm front may bring a few showers late in the day.

<output_forecast_test>:

"daily_forecast": [
    {
        "date": "20190927",
        "weekday": "Friday",
        "forecast": "A high pressure ridge over New England will bring mainly fair and cool weather tonight."
    },
    {
        "date": "20190928",
        "weekday": "Saturday",
        "forecast": "A warm front followed by a cold front will bring a threat of showers and thunderstorms."
    },
    {
        "date": "20190929",
        "weekday": "Sunday",
        "forecast": "High pressure building in from Ontario will bring fair and cool weather."
    },
    {
        "date": "20190930",
        "weekday": "Monday",
        "forecast": "Monday will start out fair, but an approaching warm front may bring a few showers late in
        ↪  the day."
    }
```

### F.2. Report Aspect and Claim Extraction

The Instruction prompt for extracting discussion aspect and claim from the report is provided as follows:

## Report Aspect and Claim Extraction

```
You are an advanced weather forecast text analysis model.
Your task is to analyze the meaning of the forecast texts semantically and
classify each day's forecast into the most relevant meteorological categories
and subcategories.

The valid classification hierarchy is defined below:
## group_dict (Keyword Groups)
<keyword_dictionary>

Each subcategory contains example words or phrases. These examples serve as
semantic references, not strict matching tokens. You should classify the
forecast based on meaning.

## daily_forecast (Input Data for All Days)
<forecast_text>

---

Please output **strict JSON**, formatted as a list:
[
{{
    "date": "YYYYMMDD",
    "weekday": "xxx",
    "claims": ["high_pressure","Storm","hot_temperature","dry_air"],
    "aspects": ["Pressure_System,""Event","Temperature","Humidity"]
}},
...
]

Requirements:

- **claims**:
The detected **claims** (from `group_dict`).
Each selected subcategory must be chosen from the valid list:
<claim_category>

- **aspects**:
The parent **categories** corresponding to each subcategory.
Each category must be selected from:
<Aspect>

### RULES

1. Do **not** rely solely on exact word matching; use **semantic interpretation**.
2. If no keywords are detected for a day, return empty lists.
3. Do **not** output explanations, comments, markdown, or any text outside valid JSON.
4. Output must be valid JSON at the top level--no trailing commas.
```

| Aspect | Claim Category | Representative Keywords |
|---|---|---|
| Temperature | Hot / Warm | warming; warmer temperatures; hot temperatures; increasing temperatures; temperatures increase; above average temperatures; above normal temperatures; warm; warmer; hot; high temperatures; warmup; heat; temperatures will moderate; temperatures rebound |
| | Cool / Cold | colder; dropping temperatures; cool; frigid; cold; cooling; wintry; cooler; falling temperatures; temperatures fall; below average temperatures; below normal temperatures; plummet temperatures; chills; winter weather; freeze |
| | Moderate | normal temperatures; mild temperatures |
| Wind | Strong wind | blustery; strong winds; strong westerly winds; gusts; gusty; gusty winds; damaging winds; dangerous wind; high winds; strong west winds; strong southwest winds; stronger winds; winds will be strong; winds increasing; increasing winds; increase winds; increase in winds; increase in southwesterly winds; winds will increase; winds will rapidly increase; winds will pick up; winds will strengthen; winds to increase; winds will be on the increase; winds will also be on the increase; winds will crank back up; crank up the winds; kicking up the winds |
| | Light wind | windy; breezy; breezy to windy; weak wind; breezes; less wind; winds will decrease; winds will taper off; winds will subside; winds subside; winds will diminish |
| Humidity | Dry air | low humidity; lower humidity; dry; drier |
| | Moist air | high humidity; raising humidity; moist; damp; humid; wet |
| Frontal System | Cold front | cold front; backdoor cold front |
| | Warm front | warm front |
| Pressure System | High pressure | high pressure; the high; another high; this high |
| | Low pressure | low pressure; the low; low pressure system; that low; upper low; another low; coastal low |
| Wave Pattern | Ridge | ridge |
| | Trough | trough |
| Wind Flow System | Onshore flow | onshore flow |
| | Offshore flow | offshore flow |
| Event | Precipitation | precipitation; rain; rainfall; shower; showers; drizzle; drizzly; rain showers |
| | Snow | flurries; snow; snowfall; snows; snow shower; snow showers; hail; hails |
| | Storm | storm; storms; thunderstorm; thunderstorms; hurricane; cyclone |

*Table 13*. Standardized annotation protocol for each aspect and claim category with reference example in weather claim classification.

**Temporal Forecast Extraction Example**

```
<forecast_test>:

"daily_forecast": [
    {
        "date": "20190927",
        "weekday": "Friday",
        "forecast": "A high pressure ridge over New England will bring mainly fair and cool weather tonight."
    },
    {
        "date": "20190928",
        "weekday": "Saturday",
        "forecast": "A warm front followed by a cold front will bring a threat of showers and thunderstorms."
    },
    {
        "date": "20190929",
        "weekday": "Sunday",
        "forecast": "High pressure building in from Ontario will bring fair and cool weather."
    },
    {
        "date": "20190930",
        "weekday": "Monday",
        "forecast": "Monday will start out fair, but an approaching warm front may bring a few showers late in
        ↪   the day."
    }

<extracted_keyword>:

"daily_forecast": [
  {
    "date": "20190927",
    "weekday": "Friday",
    "forecast": "A high pressure ridge over New England will bring mainly fair and cool weather tonight.",
    "claims": ["cool_temperature", "high_pressure"],
    "aspect": ["Pressure_System","Temperature"]
  },
  {
    "date": "20190928",
    "weekday": "Saturday",
    "forecast": "A warm front followed by a cold front will bring a threat of showers and thunderstorms.",
    "claims": ["Cold_Front", "Precipitation", "Storm", "Warm_Front"],
    "aspect": ["Event", "Frontal_System"]
  },
  {
    "date": "20190929",
    "weekday": "Sunday",
    "forecast": "High pressure building in from Ontario will bring fair and cool weather.",
    "claims": ["cool_temperature", "high_pressure"],
    "aspect": ["Pressure_System", "Temperature"]
  },
  {
    "date": "20190930",
    "weekday": "Monday",
    "forecast": "Monday will start out fair, but an approaching warm front may bring a few showers late in the
    ↪   day.",
    "claims": ["Precipitation", "Warm_Front"],
    "aspect": ["Event", "Frontal_System"]
  }
]
```

## F.3. Ranking Prompts for LLM in our experiments.

**System Prompt**

```
You are an expert meteorological evaluator specializing in assessing the quality of generated weather
↪  forecasts.
Here is a weather forecasting scenario, including the ground truth forecast and eight candidate predicted
↪  forecasts.
From the perspective of a professional weather analyst, you are required to rank the quality of these
↪  responses based on the following criteria:
(1) Consistency with the meteorological facts in the ground truth forecast (short for Fact.Cons);
(2) Quality of summarization of key weather signals without introducing misleading emphasis or irrelevant
↪  detail (short for Summ.Qual).
Rubric Definitions:
- Fact.Cons evaluates whether the predicted forecast is factually consistent with the ground truth in terms of
↪  key meteorological variables (e.g., temperature, precipitation, wind, synoptic systems), temporal
↪  alignment, and physical plausibility. Hallucinated or incorrect weather events should be heavily
↪  penalized.
- Summ.Qual evaluates how well the response summarizes the key weather signals clearly and concisely, without
↪  introducing misleading emphasis, unnecessary detail, or obscuring the main forecast narrative.
To help you rank these responses, we additionally provide background information, including forecast period,
↪  key weather variables, and the reference forecast.
You should generate the response in the following format:
Fact.Cons: R9 > R2 ... > R1;
Summ.Qual: R9 > R2 ... > R1.
After the rankings, provide several sentences explaining your evaluation.
Important guidelines:
- Focus only on factual and meteorological correctness.
- Penalize hallucinated weather events or incorrect trends heavily.
- Overly verbose answers that obscure the core forecast should be ranked lower in Summ.Qual.
- Minor wording differences are acceptable if the facts are preserved.
```

**User Prompt**

```
Ground Truth Forecast:\n{ground truth report}
Candidate Forecasts:
R1: {Report1}
R2: {Report2}
R3: {Report3}
R4: {Report4}
R5: {Report5}
R6: {Report6}
R7: {Report7}
R8: {Report8}
R9: {Report9}
Please rank them following the required format.
```

## F.4. Prompt For Baseline Model

**System Prompt**

```
As an AI assistant with expertise in severe weather analysis and forecasting, you are equipped to interpret
↪  comprehensive figures that illustrate various weather variables crucial for understanding the latest
↪  weather conditions across {City}. Your responsibility as a weather forecaster is to produce a general
↪  weather forecast for the future using the current weather condition images provided.
```

**User Prompt**

```
You must generate a concise multi-day weather forecast based on a template. For each day, replace the
↪  placeholder content with a natural language summary consistent with the weather phenomena represented in
↪  the figures. Keep the description concise and focused on the keyword groups listed for each date.
↪  <Example>Below are a few examples of weather analysis to help understand how the region and type of
↪  concern relate to different weather conditions:\n <Few-Shot><\Example> <Problem><Encode Weather
↪  Parameters> <Report Discussion Point><\Problem>
```

---

**<Encode Weather Parameters>**

- **land_sea_mask**: The proportion of land as opposed to ocean or inland waters.

- **10m_u_component_of_wind**: The eastward component of the 10m wind, representing horizontal air motion toward the east.

- **10m_v_component_of_wind**: The northward component of the 10m wind, representing horizontal air motion toward the north.

- **2m_temperature**: Air temperature measured at 2 meters above the surface.

- **mean_sea_level_pressure**: Atmospheric pressure adjusted to mean sea level.

- **sea_surface_temperature**: Temperature of seawater near the surface.

- **snow_depth**: Depth of snow over snow-covered areas.

- **surface_pressure**: Atmospheric pressure at the surface.

- **total_cloud_cover**: Fraction of a grid cell covered by clouds.

- **total_precipitation_6hr**: Total precipitation accumulated over the past 6 hours.

- **total_column_water_vapour**: Vertically integrated water vapor content.

- **total_column_water**: Vertically integrated liquid water content.

**Repeat for each parameter above and append to <Encoded Parameters>:**
Encode the image parameter by appending its parameter description and its corresponding encoded image.

**<Encode Weather Parameters>**:

```
The following 12 figures represent weather conditions at <time>, and each figure contains a weather parameter.
↪   The variable in each figure is provided as follows: <Encoded Parameters>
```

---

**<Report Discussion Point>**

```
<Template><<date, weekday>> Report:\n##Focus on:<weather aspect1><weather aspect2><weather aspect3>
↪   ##\n\n<<date, weekday>> Report:\n##Focus on:<weather aspect1><weather aspect2><weather aspect3>
↪   ##\n\n<<date, weekday>> Report:\n##Focus on:<weather aspect1><weather aspect2><weather aspect3>
↪   ##\n\n<<date, weekday>> Report:\n##Focus on:<weather aspect1><weather aspect2><weather aspect3> ##\n\n##
↪   </Template>
```

---

**<Few-Shot>**

**Repeat N times for few-shot:**
Example ith input:

```
    Modify <Encode Weather Parameters> given <time> and <Encoded Parameters>, and <Report Discussion Point>
```

Example ith answer:

```
    <<date, weekday>> Report:\n <content> ##\n\n<<date, weekday>> Report:\n <content> ##\n\n<<date, weekday>>
    ↪   Report:\n <content> ##\n\n<<date, weekday>> Report:\n <content> ##\n\n
```

# G. Case Study

**Question:**

As an AI assistant with expertise in weather forecasting, you are equipped to interpret comprehensive figures illustrating various weather variables crucial for understanding the latest weather conditions across the Blacksburg, Virginia.\nYour responsibility as a weather forecaster is to provide accurate and timely insights into weather conditions.\nThe following figures represent weather conditions on 20220228_0100

(Monday).\n\image>\n\n\image>\n\n\image>\n\n\image>\n\n\image>\n\n\image>\n\n\image>\n\n\image>\n\n\image>\n\n\image>\n\n\image>\n\n\image>\n\nYou must generate a concise multi-day weather forecast.\nFollow this exact output structure for each day:\n\n<<20220228, Monday>> Report:\n##Focus on: Humidity(dry_air or moist_air), Pressure_System(high_pressure or low_pressure)##\n\n<<20220301, Tuesday>> Report:\n##Focus on: Humidity(dry_air or moist_air), Pressure_System(high_pressure or low_pressure)##\n\n<<20220302, Wednesday>> Report:\n##Focus on: Temperature(hot_temperature or cool_temperature or moderate_temperature)##\n\n<<20220303, Thursday>> Report:\n##Focus on: Frontal_System(Cold_Front or Warm_Front)##\n\nFor each day, replace the placeholder content with a natural language summary consistent with the weather phenomena represented in the figures. Keep the description concise and focused on the keyword groups listed for each date.\n

**Visual Input Example:**

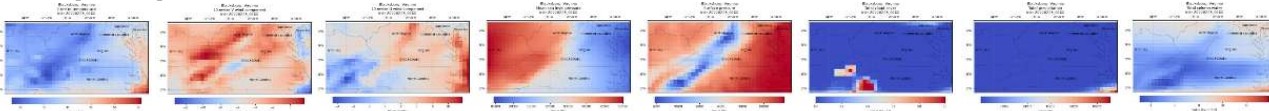

**Report:**

**Ground Truth:** <<20220228, Monday>> Report:\nDry weather, mainly with high pressure, is expected today.\n\n<<20220301, Tuesday>> Report:\nDry weather, mainly with high pressure, is expected through Tuesday.\n\n<<20220302, Wednesday>> Report:\nTemperatures will be trending warmer and will be above normal by Wednesday.\n\n<<20220303, Thursday>> Report:\nA cold front approaches the Mid Atlantic region from the north on Thursday.

**WeatherSyn:** <<20220228, Monday>> Report: High pressure will bring dry weather today.\n\n<<20220301, Tuesday>> Report: High pressure will bring dry weather to the area through Tuesday.\n\n<<20220302, Wednesday>> Report: A warming trend will continue through Wednesday.\n\n<<20220303, Thursday>> Report: A cold front will cross the area on Thursday.

**GPT-5-Nano (2-shot):** <<20220228, Monday>> Report: Dry air under a strong high-pressure system will bring quiet weather with mostly clear to partly cloudy skies and light winds.<<20220301, Tuesday>> Report: Moist air under lingering high pressure keeps skies mostly cloudy with higher humidity; no significant precipitation is expected through the day.<<20220302, Wednesday>> Report: Moderate temperatures for late February, seasonable and mild with a mix of sun and clouds and light winds.<<20220303, Thursday>> Report: A cold front crosses the region, bringing cooler air and gusty winds.

*Figure 10.* A typical case for Blacksburg, Virginia

**Question:**

As an AI assistant with expertise in weather forecasting, you are equipped to interpret comprehensive figures illustrating various weather variables crucial for understanding the latest weather conditions across the Charleston, West Virginia.\nYour responsibility as a weather forecaster is to provide accurate and timely insights into weather conditions.\nThe following figures represent weather conditions on 20220413_0200

(Wednesday).\n\image>\n\n\image>\n\n\image>\n\n\image>\n\n\image>\n\n\image>\n\n\image>\n\n\image>\n\n\image>\n\n\image>\n\nYou must generate a concise multi-day weather forecast.\nFollow this exact output structure for each day:\n\n<<20220413, Wednesday>> Report:\n##Focus on: Event(Precipitation or Snow or Storm), Frontal_System(Cold_Front or Warm_Front)##\n\n<<20220414, Thursday>> Report:\n##Focus on: Event(Precipitation or Snow or Storm), Frontal_System(Cold_Front or Warm_Front)##\n\n<<20220415, Friday>> Report:\n##Focus on: Humidity(dry_air or moist_air)##\n\n<<20220416, Saturday>> Report:\n##Focus on: No keywords##\n\nFor each day, replace the placeholder content with a natural language summary consistent with the weather phenomena represented in the figures. Keep the description concise and focused on the keyword groups listed for each date.\n

**Visual Input Example:**

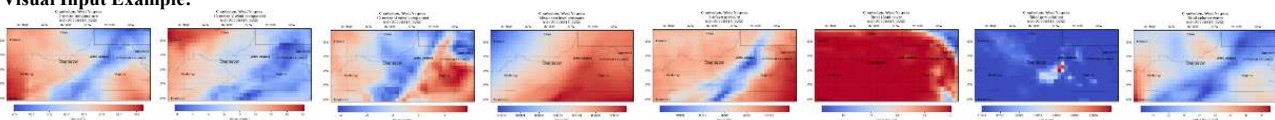

**Report:**

**Ground Truth:** <<20220413, Wednesday>> Report: A strong cold front Wednesday night brings more showers and isolated thunderstorms. <<20220414, Thursday>> Report: The strong cold front continues into Thursday morning bringing more showers and isolated thunderstorms.<<20220415, Friday>> Report: Drier for Friday.

**WeatherSyn:** <<20220413, Wednesday>> Report: A cold front will move through the area today, bringing a chance of showers and thunderstorms.<<20220414, Thursday>> Report: A cold front will bring showers and thunderstorms to the area Thursday. <<20220415, Friday>> Report: Drier weather is expected Friday.

**GPT-5-Nano (2-shot):** <<20220413, Wednesday>> Report: Showers possible as a warm front crosses the Charleston area today. Cloudy skies with temps around the 60s F. Light southerly winds.<<20220414, Thursday>> Report: A cold front moves through, bringing a band of showers and cooler air. Highs near 50s with breezy northwest winds.<<20220415, Friday>> Report: Moist air persists with higher humidity and possible isolated showers. Cloud cover typical for spring; highs in the mid-50s to mid-60s F.

*Figure 11.* A typical case for Charleston, West Virginia

**Question:**

As an AI assistant with expertise in weather forecasting, you are equipped to interpret comprehensive figures illustrating various weather variables crucial for understanding the latest weather conditions across the Great Falls, Montana.\nYour responsibility as a weather forecaster is to provide accurate and timely insights into weather conditions.\nThe following figures represent weather conditions on 20220622_0000 (Wednesday).\n<image>\n\n<image>\n\n<image>\n\n<image>\n\n<image>\n\n<image>\n\n<image>\n\n<image>\n\n<image>\n\n<image>\n\n<image>\nYou must generate a concise multi-day weather forecast.\nFollow this exact output structure for each day:\n\n<<20220622, Wednesday>> Report:\n##Focus on: Humidity(dry_air or moist_air), Pressure_System(high_pressure or low_pressure), Temperature(hot_temperature or cool_temperature or moderate_temperature)##\n\n<<20220623, Thursday>> Report:\n##Focus on: Event(Precipitation or Snow or Storm), Frontal_System(Cold_Front or Warm_Front)##\n\n<<20220624, Friday>> Report:\n##Focus on: Temperature(hot_temperature or cool_temperature or moderate_temperature)##\n\n<<20220625, Saturday>> Report:\n##Focus on: Temperature(hot_temperature or cool_temperature or moderate_temperature)##\n\nFor each day, replace the placeholder content with a natural language summary consistent with the weather phenomena represented in the figures. Keep the description concise and focused on the keyword groups listed for each date.\n

**Visual Input Example:**

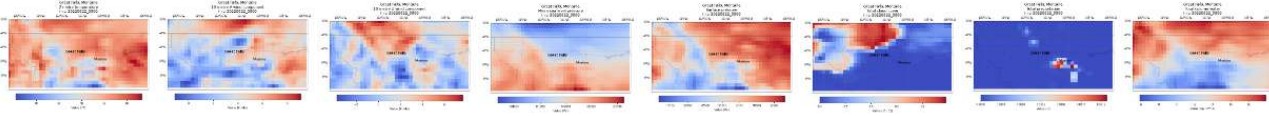

**Report:**

**Ground Truth:** <<20220622, Wednesday>> Report: Warm and mostly dry conditions are expected today as high pressure remains in control.<<20220623, Thursday>> Report: A cold front will move through Thursday afternoon and evening, which will bring a chance for a few showers and thunderstorms as it moves through.<<20220624, Friday>> Report: Cooler temperatures are expected for Friday.<<20220625, Saturday>> Report: Cooler temperatures are expected for Saturday.

**WeatherSyn:** <<20220622, Wednesday>> Report: High pressure will bring dry weather and above normal temperatures.<<20220623, Thursday>> Report: A cold front will bring a chance of showers and thunderstorms to the region on Thursday. <<20220624, Friday>> Report: Temperatures will cool back to near normal Friday.<<20220625, Saturday>> Report: Cooler temperatures will continue into Saturday.

**GPT-5-Nano (2-shot):** <<20220622, Wednesday>> Report: Moist air with a dominant high-pressure system over the region, producing moderate temperatures.<<20220623, Thursday>> Report: Light precipitation possible as a warm front slides through.<<20220624, Friday>> Report: Moderate temperatures expected with generally dry conditions.<<20220625, Saturday>> Report: Moderate temperatures continue with dry conditions.

*Figure 12.* A typical case for Great Falls, Montana

**Question:**

As an AI assistant with expertise in weather forecasting, you are equipped to interpret comprehensive figures illustrating various weather variables crucial for understanding the latest weather conditions across the Greer, South Carolina.\nYour responsibility as a weather forecaster is to provide accurate and timely insights into weather conditions.\nThe following figures represent weather conditions on 20221218_0100 (Sunday).\n<image>\n\n<image>\n\n<image>\n\n<image>\n\n<image>\n\n<image>\n\n<image>\n\n<image>\n\n<image>\n\n<image>\n\n<image>\nYou must generate a concise multi-day weather forecast.\nFollow this exact output structure for each day:\n\n<<20221218, Sunday>> Report:\n##Focus on: Humidity(dry_air or moist_air), Pressure_System(high_pressure or low_pressure), Temperature(hot_temperature or cool_temperature or moderate_temperature)##\n\n<<20221219, Monday>> Report:\n##Focus on: Humidity(dry_air or moist_air), Pressure_System(high_pressure or low_pressure), Temperature(hot_temperature or cool_temperature or moderate_temperature)##\n\n<<20221220, Tuesday>> Report:\n##Focus on: Event(Precipitation or Snow or Storm), Pressure_System(high_pressure or low_pressure)##\n\n<<20221221, Wednesday>> Report:\n##Focus on: Humidity(dry_air or moist_air)##\n\nFor each day, replace the placeholder content with a natural language summary consistent with the weather phenomena represented in the figures. Keep the description concise and focused on the keyword groups listed for each date.\n

**Visual Input Example:**

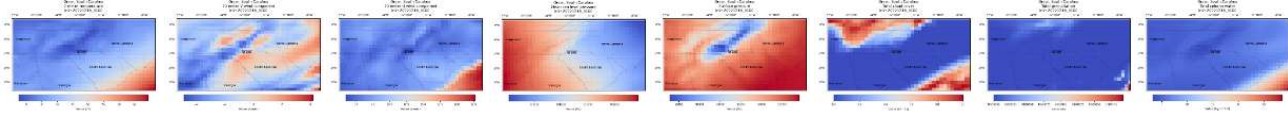

**Report:**

**Ground Truth:** <<20221218, Sunday>> Report: Dry and cool high pressure builds in from the west Sunday.<<20221219, Monday>> Report: Dry and cool high pressure builds in from the west into Monday.<<20221220, Tuesday>> Report: A weak low pressure system tracking across the Gulf brings a chance of light precipitation mainly across the southern zones on Tuesday.<<20221221, Wednesday>> Report: Brief dry conditions can be expected on Wednesday.

**WeatherSyn:** <<20221218, Sunday>> Report: Cold and dry high pressure will build over the area today.\n\n<<20221219, Monday>> Report: High pressure will build in from the west Monday, bringing dry and colder air into the region.<<20221220, Tuesday>> Report: A low pressure system will bring a chance of rain Tuesday.<<20221221, Wednesday>> Report: Drier conditions are expected Wednesday.

**GPT-5-Nano (2-shot):** <<20221218, Sunday>> Report: Moist air under a broad high-pressure ridge brings dry, mild weather for Greer today. Skies are mostly sunny with light winds; highs in the low to mid 60s F and lows in the upper 40s to mid 50s. <<20221219, Monday>> Report: Continued moist air with a stable high-pressure setup. Expect sunny to partly cloudy skies, light winds, and highs around the mid 60s F with nightly lows near the 40s to 50 F. <<20221220, Tuesday>> Report: A weak low-pressure system nudges toward the region, bringing a chance of light showers mainly in the afternoon. Temperatures stay mild, with highs around the upper 50s to around 60 F and clouds remaining common;.prepare for a few wet periods and breezy spells.\n\n<<20221221, Wednesday>> Report: Moist air continues to dominate with lingering cloudiness. Expect humid, mild conditions with highs near the low to mid 60s F and light winds; some morning fog or low clouds possible.

*Figure 13.* A typical case for Great Greer, South Carolina

**Question:**

As an AI assistant with expertise in weather forecasting, you are equipped to interpret comprehensive figures illustrating various weather variables crucial for understanding the latest weather conditions across the Las Vegas, Nevada.\nYour responsibility as a weather forecaster is to provide accurate and timely insights into weather conditions.\nThe following figures represent weather conditions on 20220401_1100
(Friday).\n\n<image>\n\n<image>\n\n<image>\n\n<image>\n\n<image>\n\n<image>\n\n<image>\n\n<image>\n\n<image>\n\n<image>\n\n<image>\nYou must generate a concise multi-day weather forecast.\nFollow this exact output structure for each day:\n\n<<20220401, Friday>> Report:\n##Focus on: No keywords##\n\n<<20220402, Saturday>> Report:\n##Focus on: Humidity(dry_air or moist_air), Temperature(hot_temperature or cool_temperature or moderate_temperature)##\n\n<<20220403, Sunday>> Report:\n##Focus on: Event(Precipitation or Snow or Storm), Temperature(hot_temperature or cool_temperature or moderate_temperature)##\n\n<<20220404, Monday>> Report:\n##Focus on: Humidity(dry_air or moist_air), Temperature(hot_temperature or cool_temperature or moderate_temperature)##\n\nFor each day, replace the placeholder content with a natural language summary consistent with the weather phenomena represented in the figures. Keep the description concise and focused on the keyword groups listed for each date.\n

**Visual Input Example:**

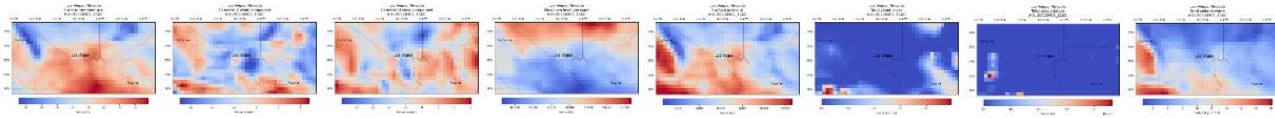

**Report:**

**Ground Truth:** <<20220402, Saturday>> Report: Dry conditions and warming temperatures will continue through Saturday.<<20220403, Sunday>> Report: The next system will briefly lower temperatures on Sunday and bring slight chances of rain showers to Lincoln County.<<20220404, Monday>> Report: Dry conditions with warmer temperatures will return Monday and Tuesday.

**WeatherSyn:** <<20220402, Saturday>> Report: Dry and warm conditions are expected.<<20220403, Sunday>> Report: A weak weather system will bring a chance of showers and cooler temperatures.<<20220404, Monday>> Report: Dry and warm conditions are expected.

**GPT-5-Nano (2-shot):** <<20220402, Saturday>> Report: Dry air with moderate temperatures; highs in the mid-70s Fahrenheit and plenty of sun.<<20220403, Sunday>> Report: Chance of light precipitation or showers; cooler temperatures, around the 60s to low 70s Fahrenheit.<<20220404, Monday>> Report: Dry air with moderate temperatures; highs in the mid to upper 70s Fahrenheit.

*Figure 14.* A typical case for Las Vegas, Nevada

**Question:**

As an AI assistant with expertise in weather forecasting, you are equipped to interpret comprehensive figures illustrating various weather variables crucial for understanding the latest weather conditions across the San Diego, California.\nYour responsibility as a weather forecaster is to provide accurate and timely insights into weather conditions.\nThe following figures represent weather conditions on 20221110_1000
(Thursday).\n\n<image>\n\n<image>\n\n<image>\n\n<image>\n\n<image>\n\n<image>\n\n<image>\n\n<image>\n\n<image>\n\n<image>\n\n<image>\nYou must generate a concise multi-day weather forecast.\nFollow this exact output structure for each day:\n\n<<20221110, Thursday>> Report:\n##Focus on: Humidity(dry_air or moist_air), wind(strong_wind or light_wind)##\n\n<<20221111, Friday>> Report:\n##Focus on: Humidity(dry_air or moist_air), wind(strong_wind or light_wind)##\n\n<<20221112, Saturday>> Report:\n##Focus on: Temperature(hot_temperature or cool_temperature or moderate_temperature), Wind_Flow_System(Onshore or Offshore)##\n\n<<20221113, Sunday>> Report:\n##Focus on: Pressure_System(high_pressure or low_pressure), Temperature(hot_temperature or cool_temperature or moderate_temperature), Wind_Flow_System(Onshore or Offshore), wind(strong_wind or light_wind)##\n\nFor each day, replace the placeholder content with a natural language summary consistent with the weather phenomena represented in the figures. Keep the description concise and focused on the keyword groups listed for each date.\n

**Visual Input Example:**

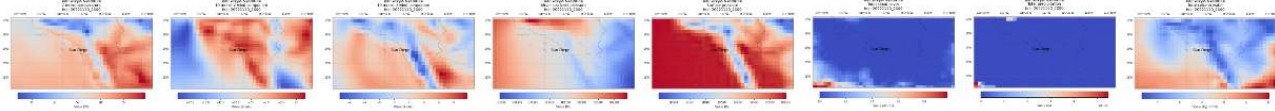

**Report:**

**Ground Truth:** <<20221110, Thursday>> Report: Dry and a little warmer with locally gusty northeast winds in the mountain passes and canyons.\n\n<<20221111, Friday>> Report: Dry and a little warmer with locally gusty northeast winds in the mountain passes and canyons.<<20221112, Saturday>> Report: Weak onshore flow will begin to spread cooling inland to the coastal mountain slopes.<<20221113, Sunday>> Report: A low pressure system moving inland to the north will bring stronger onshore flow, with lower temperatures and gusty west winds for the mountains and deserts.

**WeatherSyn:** <<20221110, Thursday>> Report: Gusty northeast winds and dry conditions are expected today.<<20221111, Friday>> Report: Gusty northeast winds and dry conditions are expected through Friday.<<20221112, Saturday>> Report: Onshore flow will increase Saturday with a cooling trend.<<20221113, Sunday>> Report:Onshore flow will increase Sunday with a cooling trend. A strong low pressure system will bring strong onshore flow and cooler temperatures Sunday night.

**GPT-5-Nano (2-shot):** <<20221110, Thursday>> Report: Onshore flow with a respectful marine layer keeps the coast damp with morning fog at times. Inland areas stay mild and dry. No significant precipitation expected.<<20221111, Friday>> Report: Moist onshore flow persists; coastal clouds and fog are possible overnight into morning. Otherwise, partly sunny and mild inland with little to no rain.<<20221112, Saturday>> Report: Coastal temperatures near seasonal norms; onshore breeze keeps inland temperatures comfortable. Mostly sunny to partly cloudy with minimal precipitation.<<20221113, Sunday>> Report: Weak low-pressure influence brings cooler, cloudier conditions; onshore flow maintains coastal clouds. Light winds with a slight chance of a light shower; otherwise dry.

*Figure 15.* A typical case for San Diego, California

