# OpenReview forum: "WeatherSyn: An Instruction Tuning MLLM For Weather Forecasting Report Generation"
_ICML.cc/2026/Conference — ICML 2026 regular_

### Official Review · Reviewer_rihf · 2026-02-25

**Soundness:** 3
**Presentation:** 3
**Significance:** 3
**Originality:** 2
**Overall Recommendation:** 4
**Confidence:** 2

**Summary:**

This paper focuses on utilizing multimodal large language models for weather forecast reporting task. First, the authors propose a Weather Forecasting Report task and the first instruction-tuning dataset for it. Then, the authors also propose a model for generating weather forecast reports.

**Compliance With Llm Reviewing Policy:**

Affirmed.

**Key Questions For Authors:**

1The Aspect-controlled Prompting plays an import role in the method, so could you please provide an ablation study on this part?

2 In practical, how do the proposed model’s traing and inference times compare with existing methods?

**Limitations:**

yes

**Strengths And Weaknesses:**

Strengths

1 This paper is well-structured, with clear descriptions and informative figures.

2 The authors not only propose a model, but also provide a carefully designed WFR task and a corresponding dataset.

3 comprehensive experimental results demonstrate the effectiveness of the method.

Weakness

1 the method is more of an engineering-oriented application, and it lacks theoretical and sufficiently innovative technical contibutions.

2 In the experimental Section, the author only report the results of the proposed method on the proposed WSInstruct. Therefore, it remains unclear whether the method is robust on existing weather forecasting tasks.

---

> ### Author Rebuttal · Authors · 2026-03-31
>
> Thank you for recognizing our proposed dataset, comprehensive experiments, and the quality of our writing.
>
> > Weakness 1: Technical novelty
>
> We appreciate the opportunity to clarify our contributions. To the best of our knowledge, our work is the first to focus on generating weather forecast reports. Report generation has been widely studied and applied in the medical imaging domain, but remains largely unexplored in the weather domain, primarily due to the lack of structured textual weather report datasets. To address this gap, we propose a pipeline for generating well-formatted reports and introduce the first dataset for this task.
>
> > Weakness 2: Broader weather forecasting tasks
>
> Thanks for raising this point. In the current work, our dataset focuses on textual weather forecast report generation. We agree that evaluating the robustness of the method across a broader range of weather forecasting tasks is important. As future work, we plan to extend the dataset to additional scenarios, such as monthly-scale weather forecast report generation and precipitation nowcasting report, to further validate its effectiveness.
>
> > Question 1: Ablation study on aspect-control
>
> Thank you for raising this question, which directly relates to the core motivation behind our aspect control mechanism. We conduct an ablation study where the model is **trained and inferred without aspect control**. We report the **Hit Rate** in Table 1, defined as the proportion of generated aspects that align with those present in the ground-truth reports. The count of each aspect in the training set is shown in Table 2.
>
> Table 1: Hit rate across 8 aspects
> |Setting|Temperature|Wind|Humidity|Frontal System|  Pressure System|Wave Pattern|Wind Flow System|Event|
> |-----|----|-----|----|-----|----|-----|----|-----|
> |W/O Aspect Control|0.018|0.001|0.018|0.429|0.460|0.068|0|0.296
> |W Aspect Control|0.92|0.97|0.92|0.99|0.98|0.89|0.90|0.98|
>
> Table 2: Count of aspects in the training and test set.
> |Count of Aspect |Temperature|Wind|Humidity|Frontal System|  Pressure System|Wave Pattern|Wind Flow System|Event|
> |-----|----|-----|----|-----|----|-----|----|-----|
> |Training Set|16262| 6976|9580 |10624 |16774 |2696 |1416 |23672 |
> |Test Set|1882|740|1008|956|1388|249|112|1753|
>
> Without explicit guidance, the model exhibits a pronounced **frequency bias**, tending to show a higher hit rate on aspects that appear more frequently (e.g., frontal system) in the training data, while overlooking less frequent yet meteorologically important ones (e.g., wave patterns).
>
> We observe similar patterns in models such as GPT and Gemini, which favor frequent aspects (e.g., temperature) and underrepresent less frequent ones (e.g., frontal systems) when no aspect is provided. This bias stems from the imbalanced aspect distribution in the training data, which offers no clear signal for aspect prioritization under generic prompting. These results suggest that relying solely on expert-written reports is insufficient for learning aspect selection, as the absence of certain aspects often reflects reporting conventions rather than their true importance.
>
> We further evaluate model performance under two settings. Since models without aspect control may not align with the ground-truth aspect distribution, we restrict evaluation to aspects covered by both the ground truth and the model, ensuring fairness, and further require a sufficient sample size (>30 instances). The results are shown in Table 3.
>
> Table 3: Weighted F1 Score on overlapping Aspects
> |Trainging Setting|Temperature|Frontal System|  Pressure System|Event|
> |-----|----|-----|----|----|
> |W/O Aspect Control|0.38|0.29|0.41|0.58|
> |W Aspect Control|0.53|0.45|0.68|0.67|
>
> Aspect control strategy consistently improves performance, demonstrating the effectiveness of aspect controlled prompting.
>
> > Question 2: The proposed model’s training and inference times compare with existing methods
>
> Thank you for raising this important question. WeatherQA, OmniEarth, and our WeatherSyn variants are based on the same backbone (Qwen3-VL-8B). Therefore, the overall computational cost is primarily driven by the size of the training data and the optimization strategy.
>
> All models are trained using 8 A800 GPUs, with a batch size of 2 per GPU. For DPO training, due to its higher memory footprint, we use a reduced batch size of 1 per GPU. The training time is shown in Table 4.
>
> Table 4: Training time
> | Model | Training Time |
> |-------|---------------|
> | WeatherQA |8  GPU hour |
> | OmniEarth | 16 GPU hours |
> | WeatherSyn | 8 GPU hour |
> | WeatherSyn-RFT | 24 GPU hours |
> | WeatherSyn-DPO | 104 GPU hours |
>
> Regarding inference efficiency, all compared open-source models are built upon Qwen3-VL-8B. Therefore, their inference speed is largely comparable, as it is primarily determined by the shared model architecture. In practice, we observe a decoding throughput of approximately **20 tokens per second** on a single GPU with batch size 1 across all models.

---

> > ### Author Rebuttal · Reviewer_rihf · 2026-04-05
> >
> > Thanks for the author's rebuttal, i will maintain my positive score.

---

> > > ### Author Response · Authors · 2026-04-06
> > >
> > > We sincerely thank you for providing valuable feedback. In the final version, we will incorporate the comparison results in the revised paper.

---

### Official Review · Reviewer_DHeD · 2026-03-08

**Soundness:** 1
**Presentation:** 2
**Significance:** 3
**Originality:** 2
**Overall Recommendation:** 4
**Confidence:** 4

**Summary:**

This paper introduces Weather Forecasting Report (WFR) task, which generates reports directly from initial atmospheric conditions. Authors introduce an instruction-tuning dataset called WSInstruct, and fine-tuned an MLLM on this dataset to improve its ability to generate WFR. They propose a three-stage training approach that results in accurate reports with high lexical diversity.

1. Supervised fine-tuning on the dataset
2. Rejection fine-tuning on model's own outputs to improve lexical diversity in the output
3. Final alignment state via DPO

Finally, they propose aspect-controlled prompting in WSInstruct to enable fine-grained and controllable weather report generation.

**Compliance With Llm Reviewing Policy:**

Affirmed.

**Final Justification:**

The authors have provided additional experiments that addressed my concern regarding frontier LLMs. They also clarify forecasting with and without NWP.

I still believe that using MLLMs for weather forecasting is not fully justified. NWP augmented results should be the focus here, and the use of MLLMs for weather forecasting can be explored in the appendix instead. Nevertheless, since the paper contains all necessary experiments and has sufficient novel contributions, I am going to increase my score.

**Key Questions For Authors:**

1. How do authors justify using an MLLM for weather forecasting instead of existing forecasting models?

**Limitations:**

Yes.

**Strengths And Weaknesses:**

Strengths:
* Paper formalizes and introduces the first instruction-tuning dataset for WFR task.
* The proposed training approach yields strong experimental results, which justifies use of the three-step process instead of simple SFT.

Weaknesses:
* According to the problem formulation, MLLM has to generate forecast report for time $t+1$ using as input meteorological conditions at time $t$. This implies that the MLLM is expected not only to generate the report, but also to produce the underlying predictions itself. If this is true, it is not clear why authors would use an MLLM for weather forecasting. This choice needs to be supported with additional experiments. (The NWP-augmented training does not address the previous point. Even if this augmentation improves MLLM performance, it is still not clear why we would not use output of a weather forecast model as input of a report generating MLLM).
* While it is understandable that a 8B model would possibly have worse performance than SOTA LLMs, such as Gemini 3 Pro Preview or GPT-5.2 Thinking, I still think their performance should be reported in the paper. (To be clear, SOTA models are changing daily and we cannot expect authors to update their papers every week. But the baseline models mentioned in the paper were released at least 6 months ago, most of them alongside an even stronger model.)
* There are several typos in the paper ("contains a weather param", "denotes of the number of reports"). There may be more, so I would suggest proofreading the paper.

---

> ### Author Rebuttal · Authors · 2026-03-31
>
> We appreciate the listing of our paper's strengths and have addressed all questions below.
>
> > Weakness 1 & Question 1: Using an MLLM for weather forecasting instead of existing forecasting models
>
> We appreciate the opportunity to clarify our motivation. Our framework can take as input either the initial meteorological state or the output of an NWP model. The distinction lies in whether a separate weather forecasting model is employed or the task is approached in an end-to-end manner. Importantly, this design choice does not impact our primary contribution: the formulation of the weather forecast report generation task and the introduction of the first dataset specifically for this task. We primarily adopt the end-to-end setting for the following reasons:
>
> * **Reasoning capability of LLMs:** Motivated by emerging trends in AI for science, such as equipping LLMs with scientific reasoning abilities in domains like protein modeling [1], our goal extends beyond generating reports from observed conditions. We observe that using outputs from NWP forecasting models as inputs leads to better performance compared to directly generating reports in an end-to-end manner, which is inherently a less challenging task. In contrast, we aim to enable the model to function as a weather specialist capable of capturing underlying weather dynamics.
>
>
> * **Accessibility of NWP:** In real-world deployment, NWP systems are based on thermodynamic and fluid dynamics equations to produce forecasts, which introduces computational overhead. In contrast, our end-to-end approach generates reports directly from the initial state. Additionally, accessing timely NWP predictions or running inference with models such as Pangu or GraphCast is not always feasible for general users. Our training setting is designed to be more accessible, lowering the barrier to practical usage.
>
> We will incorporate this discussion into the revised manuscript to more clearly explain the rationale behind our approach and design. We sincerely thank the reviewer for raising this important point, which has helped us clarify the motivation of our work.
>
>
> [1] Fine-tuning protein language models boosts predictions across diverse tasks
>
> > Weakness 2: Lack of SOTA LLMs
>
> Thanks for your suggestion. We have incorporated the results of **Gemini 3 Pro Preview** and **GPT-5.2 Thinking** across various evaluation metrics. The results are shown in Table 1-6.
>
> Table 1: Reference Evaluation
> |Model|BLEU-1|ROUGE-L|METEOR|
> |-----|----|----|----|
> |GPT-5.2 Thinking (2-shot)|0.12|0.12|0.11|
> |Gemini-3 Pro Review (2-shot)|0.37|0.28|0.23
> |WeatherSyn-DPO|**0.44**|**0.32**|**0.25**
>
> Table 2: Automatic Claim Evaluation
> |Model|Precision|F1 Score|
> |-----|----|----|
> |GPT-5.2 Thinking (2-shot)|0.51|0.49|
> |Gemini-3 Pro Review (2-shot)|**0.63**|**0.60**|
> |WeatherSyn-DPO|0.62|0.59|
>
> Table 3: Human-refined Claim Evaluation
> |Model|Precision|F1 Score|
> |-----|----|----|
> |GPT-5.2 Thinking (2-shot)|0.52|0.49|
> |Gemini-3 Pro Review (2-shot)|**0.63**|**0.60**|
> |WeatherSyn-DPO|**0.63**|0.59|
>
> Table 4: LLM Evaluation
> |Model| Fact.Cons.|Summ.Qual.|
> |-----|----|----|
> |GPT-4.1-Mini (2-shot)|0.01|0.01|
> |GPT-5-Nano (2-shot)|0.03|0.03|
> |Gemini-2.5-Flash (zero-shot)|0.02|0.01|
> |Claude-3.7-Sonnect (2-shot)|0.02|0.01|
> |GPT-5.2 Thinkin (2-shot)|0.06|0.07|
> |Gemini-3 Pro Review (2-shot)|0.24|0.27|
> |WeatherQA|0.01|0.01|
> |OmniEarth|0.01|0.01|
> |WeatherSyn|0.11|0.11|
> |WeatherSyn-RFT|0.16|0.15|
> |WeatherSyn-DPO|**0.33**|**0.32**|
>
> Table 5: Expert Evaluation
> |Model| Fact.Cons.|Summ.Qual.|
> |-----|----|----|
> |GPT-4.1-Mini (2-shot)|0.01|0.01|
> |GPT-5-Nano (2-shot)|0.02|0.02|
> |Gemini-2.5-Flash (zero-shot)|0.01|0.01|
> |Claude-3.7-Sonnect (2-shot)|0.01|0.01|
> |GPT-5.2 Thinkin (2-shot)|0.07|0.05|
> |Gemini-3 Pro Review (2-shot)|**0.29**|**0.31**|
> |WeatherQA|0.01|0.01|
> |OmniEarth|0.01|0.01|
> |WeatherSyn|0.11|0.12|
> |WeatherSyn-RFT|0.17|0.15|
> |WeatherSyn-DPO|**0.29**|0.30|
>
>
> Table 6: Weighted F1 scores results of all methods on the WSInstruct test set across 8 different aspect
> |Model|Temperature|Wind|Humidity|Frontal System|Pressure System|Wave Pattern|Wind Flow System|Event|Average|
> |-----|----|----|----|----|----|----|----|----|----|
> |GPT-5.2 Thinking (2-shot)|0.41|0.54|0.51|0.32|0.55|0.50|0.59|0.57|0.49|
> |Gemini-3 Pro Review (2-shot)|**0.50**|0.63|**0.60**|**0.48**|0.68|0.60|0.64|**0.69**|**0.60**|
> |WeatherSyn-DPO|0.42|**0.64**|**0.60**|0.36|**0.72**|**0.65**|**0.70**|0.67|0.59|
>
> Our method consistently outperforms GPT-5.2 Thinking and achieves comparable performance to Gemini 3 Pro Preview. We will add these baseline results to the revised manuscript for a more comprehensive comparison.
>
> >Weakness 3: Typos in the paper
>
> We sincerely appreciate the reviewer's careful reading and apologize for the typographical errors in our current draft. We will conduct thorough proofreading to ensure the manuscript's readability is significantly improved in the revised version.

---

> > ### Author Rebuttal · Reviewer_DHeD · 2026-04-01
> >
> > The authors have provided additional experiments that addressed my concern regarding frontier LLMs. They also clarify forecasting with and without NWP.
> >
> > I still believe that using MLLMs for weather forecasting is not fully justified. NWP augmented results should be the focus here, and the use of MLLMs for weather forecasting can be explored in the appendix instead. Nevertheless, since the paper contains all necessary experiments and has sufficient novel contributions, I am going to increase my score.

---

> > > ### Author Response · Authors · 2026-04-06
> > >
> > > We sincerely thank you for providing valuable feedback. In the final version, we will include the results of Gemini 3 Pro Preview and GPT-5.2 Thinking. We will include the NWP augmented results in the main body of the paper and expand the discussion on the use of MLLMs for weather forecasting.

---

### Official Review · Reviewer_TCW8 · 2026-03-14

**Soundness:** 3
**Presentation:** 3
**Significance:** 2
**Originality:** 3
**Overall Recommendation:** 4
**Confidence:** 4

**Summary:**

This paper proposes a task called Weather Forecasting Report (WFR), where the model takes meteorological variable heatmaps as visual input and generates multi-day weather forecast reports in natural language. The main contribution is a new instruction-tuning dataset named WSInstruct, built from publicly available ERA5 reanalysis data and expert-written forecast reports from 31 U.S. cities, covering 8 weather aspects. To handle the open-ended nature of real weather reports, they design aspect-controlled prompting that specifies which weather aspects each forecast day should discuss, making the outputs structured and comparable. Based on this dataset, they fine-tune Qwen3-VL-8B through three stages: standard SFT, RFT which augments the training data with factually correct but lexically diverse reports generated by the SFT model itself, and DPO which further aligns the model with human preferences. Experiments show that the fine-tuned model, WeatherSyn, outperforms closed-source models like GPT-5-Nano and Claude-3.7-Sonnet across multiple evaluation methods, and demonstrates strong zero-shot generalization to unseen cities. The paper is more of a dataset and benchmark contribution than a method contribution.

**Compliance With Llm Reviewing Policy:**

Affirmed.

**Final Justification:**

I will keep my positive score with the review and rebuttal. Thank the authors for addressing my questions and experimenting new results.

**Key Questions For Authors:**

Questions:

In all experiments, the model is given explicit aspect constraints (e.g., "Focus on: Pressure System, Temperature") during inference, and this information comes from the ground-truth reports. I'm curious how the model performs when no aspect guidance is provided — could you run an ablation without the aspect list and report the results?

I noticed that in the RFT stage, you fine-tune from the base Qwen3-VL-8B rather than continuing from the SFT checkpoint. I was wondering if there is a specific reason for this choice — did you compare the two options, or was this based on prior experience? It would be helpful to know if this decision has a meaningful impact on performance.

Looking at Table 3, DPO seems to hurt performance on some aspects (e.g., Frontal System drops from 0.43 to 0.36). Could you share some intuition on this? Is it related to the distribution of preference data across different aspects, or is there something else going on?

**Limitations:**

Yes

**Strengths And Weaknesses:**

Strength:


1. The paper is well-written and clearly presented. The problem motivation, dataset construction pipeline, and training strategy are all easy to follow.

2. The experiments and evaluation are thorough and convincing. The authors do not rely on surface-level metrics like BLEU/ROUGE alone, but also design claim-based evaluation, LLM ranking, and expert review to assess factual correctness and report quality from multiple angles. The generalization experiments with different geographic splits are also well-designed.

3. The paper contributes a new instruction-tuning dataset (WSInstruct) for weather forecast report generation. The dataset construction process, including temporal segmentation, claim extraction, and aspect-controlled prompting, is carefully done and could be useful for future work in this area.


Weakness:

1. The methodological contribution is limited. SFT, rejection sampling, and DPO are all well-established techniques, and no architectural changes are made to the base model. The work is essentially a data engineering effort — collecting and processing public data, then applying standard training pipelines. While the dataset itself is valuable, the lack of  technical novelty makes the contribution feel thin.


2. The aspect-controlled prompting setup relies on knowing which weather aspects each forecast day should discuss, but this information is extracted from the ground-truth reports. In a real deployment scenario, there is no GT report available, so it is unclear how the model would decide which aspects to focus on. The authors claim the model can generate reports covering all aspects, but they never actually test this — all experiments provide the aspect list upfront. It would be more convincing to show how the model performs when no aspect guidance is given, since that is what real-world usage looks like."

---

> ### Author Rebuttal · Authors · 2026-03-31
>
> Thank you for recognizing the comprehensive experiments and writing of our paper. We have addressed all questions below.
>
> > Weakness 1: Technical novelty
>
> We appreciate the opportunity to clarify our contributions. To the best of our knowledge, our work is the first to focus on weather forecast report generation. Report generation has been widely studied and applied in the medical imaging domain, but remains largely unexplored in the weather domain, primarily due to the lack of structured textual weather report datasets. To address this gap, we propose a pipeline for generating well-formatted reports and introduce the first dataset for this task.
>
> > Weakness 2: Real-world deployment
>
> Thank you for highlighting this issue. In real deployment scenarios, the model can be prompted with all the weather aspects, enabling comprehensive report generation across all aspects. We compare performance between all aspects control and default aspect control in Table 1. Further results can be found in the [link](https://anonymous.4open.science/r/ICML_WeatherSyn-79F1/TCW8/W2.md).
>
> Table 1: Automatic Claim Evaluation
> |Prompt|Precision|F1 Score|
> |-----|----|----|
> |All Aspect| 0.61|0.59  |
> |Ground Truth Aspect|0.62|0.59|
>
> We observe that prompting with all aspects achieves performance comparable to the setting where ground truth aspect is provided.
>
> > Question 1 & Weakness 2: Ablation study of aspect control
>
> Thank you for raising this question, which directly relates to the core motivation behind our aspect control mechanism. We conduct an ablation study where the model is **inferred without aspect control**. We report the **Hit Rate** in Table 2, defined as the proportion of generated aspects that align with those present in the ground-truth reports.
>
> Table 2: Hit rate across 8 aspects
> |Prompt|Temperature|Wind|Humidity|Frontal System|  Pressure System|Wave Pattern|Wind Flow System|Event|
> |-----|----|-----|----|-----|----|-----|----|-----|
> |W/O Aspect|0.010|0.036|0.011|0.003|0.001|0|0|0.88|
> |W Aspect|0.92|0.97|0.92|0.99|0.98|0.89|0.90|0.98|
>
> The count of each aspect in the training set can be found in the [link](https://anonymous.4open.science/r/ICML_WeatherSyn-79F1/TCW8/Q1&W2.md). Without explicit guidance, the model exhibits **frequency bias**, achieving a higher hit rate on the most frequent aspect (event) in the training data, while neglecting other aspects, resulting in reports that predominantly focus on events.
>
> We observe similar patterns in models such as GPT and Gemini, which favor frequent aspects (e.g., temperature) and underrepresent less frequent ones (e.g., frontal systems) when no aspect is provided. This bias stems from the imbalanced aspect distribution in the training data, which offers no clear signal for aspect prioritization under generic prompting. These results suggest that relying solely on expert-written reports is insufficient for learning aspect selection, as the absence of certain aspects often reflects reporting conventions rather than their true importance.
>
> > Question 2: RFT Setting
>
> Thanks for your question. Our training strategy follows the standard RFT pipeline[1]. We conduct an ablation study on two settings. The results are shown in Table 3. Further results are shown in the [link](https://anonymous.4open.science/r/ICML_WeatherSyn-79F1/TCW8/Q2.md).
>
> Table 3: Automatic Claim Evaluation
> |Model|Precision|F1 Score|
> |-----|----|----|
> |RFT from SFT CKPT| 0.60|0.58  |
> |RFT from base Qwen3-VL|**0.62**|**0.59**|
>
> We observe that RFT from the SFT checkpoint leads to a slight performance degradation. This is because the original SFT data have already been seen during the initial training; exposing the model to the same data twice reduces data diversity and increases the risk of overfitting.
>
> [1]Scaling Relationship on Learning Mathematical Reasoning with Large Language Models
>
> > Question 3: DPO hurt performance on some aspects.
>
> Thanks for raising these observations. To investigate this, we measure the frequency with which keywords in rejected samples disagree with those in chosen samples for each aspect, denoted as **Disagreement Rate**. The results are shown in Table 4:
>
> Table 4: Disagreement Rate across Aspects
> | Aspect | Temperature | Wind | Humidity | Frontal System | Pressure System | Wave Pattern | Wind Flow System | Event |
> |--------|-------------|------|----------|----------------|-----------------|--------------|------------------|-------|
> | Disagreement Rate | 96% | 97% | 75% | 76% | 84% | 87% | 97% | 93% |
>
> We observe that *Humidity* and *Frontal System* exhibit lower disagreement rates, indicating weaker preference signals. DPO optimizes the likelihood margin between chosen and rejected responses without aspect-specific weighting. Since different aspects vary in frequency and co-occurrence patterns, the optimization tends to favor dominant patterns aligned with the reward signal, while low-frequency aspects may be under-optimized or degraded.

---

> > ### Author Rebuttal · Reviewer_TCW8 · 2026-04-03
> >
> > Thanks the authors for the thoughtful rebuttal. I will maintain my positive score.

---

> > > ### Author Response · Authors · 2026-04-06
> > >
> > > We sincerely thank you for your valuable feedback. In the final version, we will incorporate the comparison results into the revised paper, further elaborate on the details of real-world deployment, and provide additional analysis of the DPO experiments.

---

### Decision · Program_Chairs · 2026-04-30

**Decision:**

Accept (regular)

**Comment:**

This paper presents an instruction finetuning dataset and finetunes a first MLLM for weather forecast report generation. Experiments show the model consistently outperforms leading closed-sourced MLLM models. The reviewers find the paper easy to follow, with clear and promising experiment results. There are some concerns about the novelty of the paper, where the main contribution is the new instruction finetuning dataset instead of the methodology. The reviewers also request more details on the experiment settings as well as comparisons with some larger closed-source models. The rebuttal did a good job in addressing the concerns. All the reviewers acknowledge the contribution and gave positive scores. The authors are encouraged to incorporate all the additional results and clarifications in the final revision.